

# Species Correlation Measurements in Turbulent Flare Plumes: Considerations for Field Measurements

Scott P. Seymour, Matthew R. Johnson

Mechanical & Aerospace Engineering, Carleton University, Ottawa, K1S 5B6, Canada

*Correspondence to*: Matthew R. Johnson (matthew.johnson@carleton.com)

## Abstract

Field measurement of flare emissions in turbulent flare plumes is an important and complex challenge. The simplest approaches necessarily assume that combustion species are spatially and/or temporally correlated in the plume, such that simple species ratios can be used to close a carbon balance to calculate species emission rates (*i.e.* emission factors) and flare

conversion efficiency. This study examines the veracity of this assumption and the associated implications for measurement uncertainty. A novel tunable diode laser absorption spectroscopy (TDLAS) system is used to measure the correlation between $H_2O$ and black carbon (BC) volume fractions in the plumes of a vertical, turbulent, non-premixed, buoyancy-driven lab-scale gas flare. Experiments reveal that instantaneous, path-averaged concentrations of BC and $H_2O$ can vary independently and are not necessarily well-correlated over short time intervals. The scatter in the BC/$H_2O$ ratio along a path through the plume

was well beyond that which could be attributed to measurement uncertainty and was asymmetrically distributed about the mean. Consistent with previous field observations, this positive skewness toward higher BC/$H_2O$ ratios implies short, localized, and infrequent bursts of high BC production, that are not well-correlated with $H_2O$. This demonstrates that the common assumption of fixed species ratios is not universally valid, and measurements based on limited samples, short sampling times, and/or limited spatial coverage of the plume could be subject to potentially large added uncertainty. For BC

emission measurements, the positive skewness of the BC/$H_2O$ ratio also suggests that results from small numbers of samples are more likely to be biased low. However, a bootstrap analysis of the results shows how these issues should be easily avoidable with sufficient sample size and provides initial guidance for creating sampling protocols for future field measurements using analogous path-averaged techniques.

## 1   Introduction

Flaring in the upstream oil and gas industry (UOG) is a process used to destroy unwanted combustible gas, typically in buoyancy-driven, turbulent diffusion flames atop vertical stacks or in refractory-lined pits that are open to atmosphere. In 2019, global gas flaring volumes were estimated to be 150 billion $m^3$, up 3% from 2018, reaching levels not seen since 2009 (World Bank, 2020). Flaring is preferable to venting since the 20/100-year global warming potential (GWP) of methane, a common constituent of flare gas, is 96/34 times higher than the $CO_2$ produced by flaring (Gasser et al., 2017). However,





flaring can also produce unwanted pollutants such as soot (primarily black carbon, which has its own important climate impacts), CO, oxides of nitrogen (NOx), volatile organic compounds (VOCs), and uncombusted flare gases.

Of the pollutants produced by gas flaring, black carbon (BC; the carbonaceous, strongly light-absorbing component of soot) has been suggested as the second most important atmospheric pollutant for climate forcing following $CO_2$ (*e.g.* Bond et al.,

2013; Jacobson, 2010; Ramanathan and Carmichael, 2008). Atmospheric BC directly warms the atmosphere as it is a strong absorber of both in- and out-going radiation at all wavelengths (U.S. EPA, 2012). BC that is deposited onto Arctic snow and ice has the additional effect of decreasing surface albedo (*i.e.* reflectance) and accelerating ice melting (Flanner, 2013; Sand et al., 2016). Black carbon has a relatively short atmospheric lifetime (typically between 4-12 days, *e.g.* Cape et al., 2012) and represents a unique mitigation opportunity to obtain near-term climatic benefits.


Field measurements of black carbon, and other pollutant emissions, from gas flaring in the upstream oil and gas industry are limited. The lack of direct measurements is due to the difficulties in quantifying emissions in a turbulent, inhomogeneous, and unconfined plume which freely entrains air as it evolves downstream. Field measurements generally fall into two categories, intrusive (extractive) or non-intrusive (remote) sampling. Both measurement types tend to characterize flare

performance in broad terms such as pollutant emission factors (EFs), carbon conversion efficiency (CCE) also referred to as "combustion efficiency" (CE), or destruction removal efficiency (DRE). These performance metrics are infrequently measured and often applied to a large number of flares throughout the industry. It is therefore critical to obtain accurate estimates of flare performance from field measurements. Uncertainties in global pollutant emission inventories stemming from uncertainties in the few available measurements can have significant impact on climate modeling studies (Klimont et al., 2017;

Winiger et al., 2019; Zhao et al., 2011).

Extractive sampling techniques use an aspirated sampling system to pull gas samples into an instrument suite capable of measuring the constituent species (sometimes simply $CO_2$, black carbon, and methane). The relative species concentrations are then used to close a molar- or mass-balance of carbon to determine the desired flare performance metrics (*e.g.* Gvakharia

et al., 2017; Herndon et al., 2012; Pohl et al., 1986; Strosher, 2000; Weyant et al., 2016). Non-intrusive techniques typically employ Fourier Transform Infrared Spectroscopy (FTIR, both active and passive) to measure column concentrations of each species along a line-of-sight (LOS) through the plume (*e.g.* Blackwood, 2000; Wormhoudt et al., 2012). The relative column concentrations of combustion species are similarly used to close a carbon balance. Critically, these techniques all rely on the assumption that the species in these instantaneous measurements are well-mixed, and therefore well-correlated, in the plume.

Phrased in another way, these techniques assume that a ratio of two combustion-derived species is relatively constant spatially and temporally within the plume. The veracity of this assumption is a necessary condition to obtaining representative flare performance metrics. If, for example, concentrations of carbon species are weakly or perhaps inversely correlated in time and/or space (*i.e.* a ratio of these species is not constant) within the turbulent plume, then instantaneous measurements using



the above approaches would be prone to error. Although it is possible that local variations in combustion efficiency or pollutant emission may be averaged out in some of the available field measurement techniques, the number of required measurements is uncertain, and to date there has not been study attempting to directly inspect this potential issue.

This paper presents a first investigation of the instantaneous correlation of combustion product species in turbulent flare plumes. A unique measurement system was designed to measure instantaneous, path-averaged species correlations of black carbon and $H_2O$ concentrations in vertical flare plumes to understand if variations in relative species concentrations, and therefore variations in species ratios, are present. The results of these experiments provide insight into how these variations could affect field measurement techniques that assume constant combustion species ratios (*i.e.* well-correlated species). Further data analysis highlights potential implications for current field measurement approaches that should be considered when designing future measurement campaigns.

## 2   Experimental Setup

### 2.1  Carleton University Flare Facility (CUFF)

Experiments were conducted at the Carleton University Flare Facility (CUFF) where custom gas mixtures characteristic of the upstream oil and gas industry can be flared. The facility is shown in schematic in Fig. 1, and has been previously discussed in Conrad and Johnson (2019). Custom combinations of C1-C4 alkanes, C2 and C3 alkenes, $CO_2$, and $N_2$ from pressurized gas cylinders and evaporated C5-C7 alkanes from a controlled evaporative mixer (CEM) were combined and sent to a vertical flare nozzle to be combusted. The gas and liquid constituents were metered using thermal mass flow controllers and Coriolis mass flow controllers, respectively.

The gas mixtures were sent to a vertical flare nozzle (with selectable internal diameters of 25.4 mm, 50.8 mm, and 76.2 mm) and combusted in a turbulent, buoyancy-driven, non-premixed flame. Experiments considered flare gas compositions, summarized in Table 1, characteristic of flaring in the Bakken oil region of North Dakota and in Ecuador as well as reference cases of pure methane and hydrogen. Prior to each experiment, the flare was run for up to 20 minutes to ensure the burner nozzle had reached a steady temperature.

A three-axis translating traverse system was outfitted with a square-channel semi-circular "hoop" that was positioned above the vertical flame. The hoop was manufactured from Invar, a nickel-iron alloy with a coefficient of thermal expansion of approximately $1.2\times10^{-6}$ $K^{-1}$ (roughly an order of magnitude less than stainless steel), which allowed rigid and stable mounting of the launch and collection optics of the black carbon and $H_2O$ measurement system. The 1.5-m long optical measurement path was centered over the burner and positioned 2 m above the burner exit plane. Fiber optic cables and detector signal cables





were run from launch and collection optics to the laser driving and computer hardware systems located remotely as shown in Fig. 1. This arrangement permitted the temperature-sensitive laser driving systems to be positioned well away from the flame.

## 2.2 Optical System

Figure 2 shows a schematic of the optical measurement system. Two distributed feedback (DFB) lasers at nominal wavelengths of 1428 nm and 1654 nm from NTT Electronics were used for $H_2O$ vapour and black carbon (BC) detection, respectively.

$H_2O$ volume fraction was measured using tunable diode laser absorption spectroscopy (TDLAS) and BC volume fraction was simultaneously measured via line-of-sight attenuation (LOSA). The radiation generated by each laser was fiber-coupled and sent through separate single-mode fiber networks which included taps for wavelength references (fiber ring resonators), gas reference cells, and laser power references. All reference components were housed in an enclosure thermally-stabilized by a thermoelectric heater/cooler unit to prevent reference signal drift. Each laser was mounted in an ILX laser mount (LMD-4984)

and controlled by Stanford Research Systems (SRS) laser diode controller (LDC500). Light from each network was coupled into a single fiber using a wavelength division multiplexer (WDM, WDM-12N-111-1428/1650 from Oz Optics) with optical circulators (6015-3-APC, Thorlabs) to prevent light leakage between the two fiber networks. The combined wavelengths were then sent in a single fiber to an open path beam collimator located on the optical hoop that launched the light through the flare plume towards the collection optics 1.5 m away. This arrangement ensured that the $H_2O$ and BC measurements were coincident

through the plume.

The collection optics were designed to maximize light capture despite any beam steering as the light transects the hot plume gases. Although beam steering through hot flare plumes should be minimal, <1 mrad according to Conrad et al. (2020), any light loss due to beam steering is necessarily interpreted as the presence of black carbon. To further minimize any potential

beam steering effects, a 0.75-m focal length plano-convex lens (Thorlabs, LA1978-C) was used to collect the light assuming the beam was steered at the mid-point of the open path. The collected beam was focused to half its collimated diameter by a set of plano-convex lenses (LA1134-C and LA1805-C, focal lengths of 60 mm and 30 mm, respectively) to ensure the beam was smaller than the detector photodiode diameter. The two-wavelength beam was split into its constituent wavelengths by a dichroic mirror (Thorlabs, DMLP1500R), passed through bandpass filters (FB1650-12 from Thorlabs, and 87-868 from

Edmund Optics), and directed onto transimpedance-amplified InGaAs detectors (Thorlabs, PDA20CS). Signals from all detectors were recorded using a 1 MHz oscilloscope (National Instruments, PXIe-5172) and processed using software written in LabVIEW.

The optical system was controlled by a National Instruments (NI) PXI base station which included the oscilloscope for reading

detector signals, a waveform generator used to sweep the laser injection current (NI, PXI-5406), and the laser driving systems. The injection current of the $H_2O$ detection laser was swept using a 200-Hz triangle waveform to allow the laser to scan back and forth across the two absorption peaks of $H_2O$ vapour centered at 7006.12 cm$^{-1}$ (1427.32 nm) and 7007.03 cm$^{-1}$



(1427.14 nm) corresponding to combinational vibrational bands of $(2\nu_2+\nu_3)$ and $(\nu_1+\nu_3)$, respectively. Each 5-ms sweep period, consisting of a forward and backward scan of the $H_2O$ peaks, was averaged to produce a single $H_2O$ spectral signal for

analysis. The laser output wavelength throughout each sweep was determined using a gas reference cell (80-cm path length, containing 2.16% $H_2O$ vapour by volume) in combination with a fiber-ring resonator (FRR) which had a free spectral range of 0.345 GHz (0.0115 cm$^{-1}$). The $H_2O$ absorption peaks observed in the reference cell served as absolute wavelength references while the optical resonance peaks of the FRR indicated the wavelength change throughout the laser sweep.

An estimate of the absorption-free intensity, necessary to quantify light attenuation during an experiment, was made before and after each experiment by placing a tube across the optical path and purging with $N_2$ gas for several minutes to remove the light absorbing species (*i.e.* ambient $H_2O$). Ambient $H_2O$ concentrations could also be measured after the tube was removed and prior to lighting the flame. Signals collected during experiments were processed by subtracting the detector's measured dark current and then normalizing by the absorption-free intensity estimate to yield the measured transmissivity and hence

absorption.

## 3    Measurement Theory

### 3.1  Black Carbon Detection – Line-Of-Sight Attenuation (LOSA)

Soot/black carbon content in the flare plume was measured via optical attenuation of the 1654-nm laser, where the exact wavelength was tuned so that it was not detectably absorbed by $H_2O$, $CO_2$, CO, or methane. Following the definition of

Baumgardner et al. (2012), this approach of inferring soot volume fraction from light absorption yields estimates of "black carbon" (BC), which might alternatively be called "equivalent BC" (Andreae and Gelencsér, 2006; Petzold et al., 2013) or "light-absorbing carbon" (Bond and Bergstrom, 2006). In the specific context of gas flares, fresh particulate emissions are statistically pure elemental carbon (*e.g.* Conrad and Johnson, 2019; Popovicheva et al., 2019; Schwarz et al., 2015); thus, "elemental carbon", "soot", "(equivalent) black carbon", and "light-absorbing carbon" can be considered as synonymous in

the present work.

The optical transmissivity observed through a flare plume with the 1654-nm laser can be directly related to the soot/BC volume fraction through,

$$\tau = \frac{I}{I_0} = exp\left(-K_{e_\lambda}\int_s\ f_v(s)ds\right) \tag{1}$$

where $K_{e_\lambda}$ [m$^{-1}$] is the BC extinction coefficient evaluated at some wavelength $\lambda$, $f_v$ [-] is the local BC volume fraction which can be integrated over the optical path $s$ [m], and the transmissivity $\tau$ [-] is calculated as a ratio of the measured (transmitted) intensity $I$ over the incident intensity $I_0$ (Coderre et al., 2011; Migliorini et al., 2011). If the BC extinction coefficient $K_{e_\lambda}$ is known at the operating wavelength, the transmissivity can be used to yield the integrated or path-averaged BC volume fraction.



From the review by Coderre et al. (2011), most rigorous soot optical property measurements in the literature have been conducted in/near the visible region with most not exceeding 1064 nm in the infrared. Among the few studies investigating soot optical properties between 1400-1700 nm, there is disagreement on $K_{e_\lambda}$ values. However, this is not a limiting concern in the present work, since any bias in assumed black carbon optical properties will affect absolute BC volume fractions but not the strength of correlation with $H_2O$. Thus, BC optical properties at the measurement wavelength of 1654 nm were assumed using complex refractive index values calculated from Chang & Charalampopoulos (1990) and a scattering-to-absorption ratio extrapolated from Migliorini et al. (2011) to yield a reference extinction coefficient of $2.6 \times 10^9$ m$^{-1}$.

### 3.2  $H_2O$ Absorption Spectra Isolation

The strongly light-absorbing black carbon will attenuate light at both laser wavelengths thereby corrupting the $H_2O$ vapour absorption spectra near 1428 nm. Given $K_{e_\lambda}$ values at the two measurement wavelengths, a simple ratio of Eq. (1) at each wavelength would be sufficient to isolate the $H_2O$ absorption spectra from the attenuation caused by black carbon. Rather than relying on the literature where there is some disagreement about $K_{e_\lambda}$ values in the near infrared, an experiment was conducted to estimate the ratio between BC extinction coefficients at 1654 nm and 1428 nm. This was accomplished by producing a heavily-sooting ethylene flare and observing attenuation magnitudes at both wavelengths (while the 1428-nm laser was tuned away from detectable $H_2O$ absorption peaks). As detailed in the appendix, the experimentally determined BC extinction coefficient ratio of $1.12 \pm 1.3 \times 10^{-4}$ fell within the range of values that could be calculated from the literature and most closely resembled the results of Krishnan et al. (2001).

### 3.3  $H_2O$ Detection – Tunable Diode Laser Absorption Spectroscopy (TDLAS)

The transmissivity of light through a light-absorbing gas medium is described by Beer-Lambert's law,

$$\tau(v) = \frac{I}{I_0} = \exp\left(-\int_s \frac{qp}{k_bT} \sigma(p, T, q, v) ds\right), \tag{2}$$

where $\tau(v)$ [-] is the optical transmissivity evaluated at some wavenumber $v$ [cm$^{-1}$], and the wavenumber is the inverse of the incident radiation wavelength $\lambda$. $\tau(v)$ is related to local gas properties where $q$ [-] is the volume fraction of the absorbing species, $p$ [Pa] is the pressure, and $T$ [K] is the gas temperature. The equation also depends on Boltzmann's constant $k_b$ [1.38064852×10$^{-19}$ (kg·cm$^2$)/(s$^2$·K)], and the absorption cross-section of the medium $\sigma(p, T, q, v)$ [cm$^2$/molecule] evaluated at some wavenumber. Equation (2) is computed by integrating over the optical path, $s$ [m]. Following the correction for black carbon attenuation, and in the absence of other light absorbers at the chosen wavelength, $\sigma(p, T, q, v)$ for the isolated $H_2O$ absorption spectra were computed using the *HI*gh-resolution *TRAN*smission molecular absorption database (HITRAN), and assuming a Lorentz profile to describe line broadening.



In the present application, the sample path is a line-of-sight (LOS) through a turbulent flare plume. This means that the integral in Eq. (2) cannot be directly evaluated without knowledge of the temperature and volume fraction distribution along the
measurement path. Instead, a technique called profile fitting, as implemented by Liu et al. (2007) in laminar flames, was employed here to deal with non-uniform gas properties along the LOS. The technique was selected since it was observed in preliminary simulations to have superior performance to other spectroscopic techniques developed for non-uniform sensing, at least in this application. Other techniques considered include integrated absorbance of two transitions (*e.g.* Farooq et al., 2008; Fotia et al., 2015; Liu et al.; Allen, 1998; Zhou et al., 2003) or multiple transitions in a linear regression solution (*e.g.*
Liu et al., 2013, 2005; Sanders et al., 2001; Zhang et al., 2016).

The profile fitting technique assumes that the unknown gas property distributions in Eq. (2) (in this case the $H_2O$ volume fraction $q$, and temperature $T$) can be described by some general form. Here, we assume column ("tophat") distributions with identical column widths for both $H_2O$ volume fraction and gas temperature as shown in Fig. 3. The column distributions are
a function of three independent unknowns to be solved (column width $w$, column temperature $T_{col}$, and column volume fraction $q_{col}$) and two constants (ambient temperature $T_{amb}$, and volume fraction $q_{amb}$) that can be directly measured. The assumed column distributions were substituted into Eq. (2) to compute theoretical absorption spectra which were compared with measured $H_2O$ absorption signals. The parameters describing the column distributions ($w$, $T_{col}$, $q_{col}$) were then iteratively refined using a Nelder-Mead downhill simplex optimization scheme to best match the measured absorption spectra. The final
column distributions were used to compute path-averaged $H_2O$ volume fractions for further analysis. Importantly, it is not critical to accurately reconstruct the spatial distribution of gas temperature and $H_2O$ volume fraction along the path; the column distribution simply allows for meaningful path-average values to be calculated from an unknown distribution. Preliminary simulations also showed the column distribution approach to be more stable and accurate, at least in this application, than the Gaussian distribution approach proposed by Grauer et al. (2018).


A critical consideration for developing an optical system capable of measuring path-averaged $H_2O$ volume fractions was the judicious selection of absorption transitions. After considering many potential transitions from the HITRAN database, a laser of 1428-nm nominal wavelength output was selected to measure two absorption peaks of $H_2O$ vapour centered at 7006.12 cm$^{-1}$ (1427.32 nm) and 7007.03 cm$^{-1}$ (1427.14 nm), related to combinational vibrational bands of ($2\nu_2+\nu_3$) and
($\nu_1+\nu_3$), respectively. These transitions were chosen because they had sufficient absorption strength to be measured in the lab, were isolated from other species transitions, and possessed the unique characteristic of having opposite, near-linear, scaling trends of peak absorption with increased temperature over the range of 300 K to 800 K. This last characteristic, related to their large difference in lower state energies, meant that temperature and concentration effects on the absorption peaks were effectively decoupled in the optimization scheme, increasing the overall accuracy of the method. Since expected flare plume
temperatures were <800 K (and most likely ≤600 K as suggested in thermocouple measurements by Poudenx et al., 2004),





these absorption transitions were ideally suited to this application. Poudenx et al.'s measurements were taken 1.4 m downstream of a 2.21-cm diameter flare with 2 m/s exit velocity. In this study, the measurements presented in Sect. 4 used flow velocities not exceeding 2 m/s and took LOS measurements at 2 m above the burner on buoyancy-driven turbulent flare plumes. Post-analysis of measurements presented in Sect. 4 confirmed that 95% of final measurements had path-averaged

temperatures <333 K and a maximum value of 384 K. Expected temperatures were thus also well below species reaction temperatures (*e.g.* adiabatic flame temperature of methane at 2224 K, thermal $NO_x$ formation above ~1600 K) (Turns, 2000; U.S. EPA, 1999) meaning no major chemical reactions were occurring at the measurement height above the flare which would otherwise complicate measurements.

Prior to commencing experiments, the capabilities of the profile fitting technique were tested using the large eddy simulation (LES) data for methane flares in crossflow described by Conrad et al. (2020). Species and temperature data along chords through the simulated flare plume were extracted and used to compute simulated line-integrated signals (including added noise) that would be sensed by a detector. These test data included near-field locations with high peak temperatures and localized pockets of combustion products as well as far-field locations with lower peak temperatures and more dispersed combustion

products. For line-of-sight measurements where peak temperatures were below 800 K, the profile fitting technique accurately estimated the path-averaged $H_2O$ volume fraction with a mean relative error of 0.3%; 95% of measurements were within ±3.2% relative error, and all were within ±6.3% relative error.

### 3.4 Experimental Acquisition and Processing

A typical pair of laser signals, following normalization by the absorption-free intensity estimates for each line, is shown in
Fig. 4 for an experiment running the Bakken flare gas mixture. The 1428-nm signal in Fig. 4a is the average of the lead and lag components of the 5-ms (200-Hz) triangle waveform sweep. The measurement from the 1654-nm laser shown in Fig. 4b was similarly averaged to allow for direct mapping of synchronous black carbon attenuation onto the 1428-nm signal. This allowed isolation of the $H_2O$ spectra (shown in Fig. 5 plotted against wavenumber as determined using the FRR and reference cell signals). Overlaid in black on Fig. 5 is the fitted theoretical absorption spectrum produced by the column distributions as
determined by the Nelder-Mead algorithm; the residual plot above demonstrates the closeness of fit. The synchronous BC signal (Fig. 4b) was time-averaged over the sweep duration to obtain a single BC volume fraction for comparison with the $H_2O$ estimate.

In the Nelder-Mead optimization scheme, a scalar sum-of-squared errors (SSE) value is produced representing the difference
between the measured and fitted path-averaged $H_2O$ absorption spectra. A purposefully restrictive SSE threshold of $\leq 10^{-4}$ was used to filter the raw experimental data. Filtering on the basis of SSE is shown in the appendix to remove spectral data that is not well-fitted to a theoretical absorption peak during the profile fitting, potentially due to beam steering events. However, while this tight tolerance ensured robustness of the presented results, it is possible that this artificially reduces the variability



and skewness of the BC/H$_2$O ratio as elaborated in the appendix. For example, if cases of less than ideal spectral fitting are
correlated with instances in which complex turbulent structures in the plume allow BC and H$_2$O fluctuations to be uncorrelated,
then the filtering could suppress part of the phenomena to be quantified. Future experiments with a faster time-resolution
system and/or additional measured absorption lines could explore this further. For now, however, the presented correlation
results likely represent a conservative lower bound on the spread and skewness of the BC/H$_2$O ratio.

## 4  Experimental Results and Discussion

Simultaneous path-averaged black carbon and H$_2$O volume fractions were recorded through plumes of flares over a range of
flow rates and burner diameters. As summarized in Table 1, two main flare gas compositions were considered that were
characteristic of flaring in the upstream oil and gas sector in the Bakken region of North Dakota (Brandt et al., 2016) and in
Ecuador (Conrad and Johnson, 2017). These compositions were selected to consider a range of higher heating values which
Conrad and Johnson (2017) suggested would produce differing sooting propensities. In addition, pure methane and pure
hydrogen tests were conducted to observe results in flares of low or zero black carbon content. Additional details of supporting
experiments to measure the black carbon extinction coefficient ratio, constrain beam steering uncertainties, and assess the
influence of measurement timescales are provided in the appendix.

Figure 6 shows example scatter plots of instantaneous path-averaged measurements of produced H$_2$O (measured H$_2$O volume
fraction minus measured ambient H$_2$O volume fraction) and BC volume fractions for two flare experiments. Although there
is a general correlation between the species, as should be expected since the production of BC necessitates the production of
H$_2$O vapour, the results show considerable scatter about the central trend. The variation is also not attributable to measurement
noise which is more than an order of magnitude less than the apparent variation. As elaborated in the appendix, the anticipated
error on the measured H$_2$O volume fraction was $\pm 3.0 \times 10^{-4}$ (relative error of $\pm 8.1\%$ at 95% confidence). For BC, the volume
fraction error is estimated to be $\pm 1.2 \times 10^{-10}$ (95% confidence). The low correlation coefficients of $R^2 = 0.50$ and 0.56 in Fig.
6 indicate that a substantial component of the variance is not explained by a simple linear model assuming perfect (time-
invariant) correlation between species. Similar plots were seen for all test conditions with coefficients ranging from 0.40 to
0.67.

In the context of mass- or molar-balance approaches relying on a fixed ratio of species in space and/or time throughout the
plume, histograms of BC/H$_2$O ratio data are plotted for each experiment in Fig. 7. Statistics for all cases, including hydrogen
and methane control tests are summarized in Table 2. As apparent in Fig. 7, all of the experiments with realistic flare gas
mixtures show similar distributions, with the BC/H$_2$O ratio varying about a single mode. Moreover, the distributions are all
positively skewed, with a tail towards intermittent, higher values of the BC/H$_2$O ratio. There is no apparent trend with flare
gas flow rate or flare nozzle diameter, suggesting that the variation in BC/H$_2$O ratio is an inherent characteristic of the turbulent





buoyant plumes. There is, however, an apparent dependence on flare gas composition with the larger heating value Ecuador flare gas mixtures producing broader variation in $BC/H_2O$ ratios ($7.0$–$9.3 \times 10^{-7}$ vs. $4.2$–$6.5 \times 10^{-7}$) than the Bakken flare measurements, as well as higher overall $BC/H_2O$ ratios ($20.0$–$24.8 \times 10^{-7}$ vs. $11.6$–$15.0 \times 10^{-7}$). Data for each of these mixtures is aggregated in Table 2 and replotted in Fig. 8, where the skewness of the distributions becomes more apparent. On a relative

basis, the coefficient of variance (mean-normalized standard deviation) of the $BC/H_2O$ ratios for the two mixtures are similar – 0.40 and 0.38, for Bakken and Ecuador mixtures, respectively. The skewness of the results aggregated by fuel type are also comparable (2.53 for Bakken and 2.30 for Ecuador). This again suggests that the uncorrelated variation in instantaneous species concentrations are most likely driven by turbulent dynamics within the heated, buoyant plume.

Prior to considering the possible implications of these results, it is first important to understand and bound the potential contributions of measurement uncertainty to the observed variability and skewness. Results of the hydrogen and methane tests were identically processed to glean the maximum amount of variability in an inferred $BC/H_2O$ ratio that could be observable from measurement uncertainty alone. For the $H_2$ flame, there could be no black carbon produced, and any attenuation variation in the BC measurement laser would be solely attributable to experimental uncertainty. Similarly, any minor contribution from

BC in the methane control tests would only lead to a conservatively high estimate of uncertainty. Together, the hydrogen and methane control tests, with adiabatic flame temperatures of 2379 K and 2224 K, bound those of the experimental flare gas mixtures (2249 K and 2250 K) and thus should more than encompass the range of temperature-driven beam steering effects and thermally-driven fluid structures within the plume. The measured variability in the $BC/H_2O$ ratio for the Bakken and Ecuador flare gas experiments exceeds the ratio variation of the control methane and hydrogen experiments by at least 3.7

times (median is a factor of 6.0).

In addition, a simple Monte Carlo (MC) simulation was performed to estimate $BC/H_2O$ ratio variation and skewness attributable to measurement uncertainty, as further detailed in the appendix. The simulation took median BC and $H_2O$ volume fraction results from Bakken and Ecuador datasets and computed ratios of the species volume fractions while including a

random, independently drawn contribution due to measurement uncertainty from distributions based on the control tests; further details are included in the appendix. Repeating this process 50,000 times produced a distribution of $BC/H_2O$ ratios that represents a worst-case error bound since any correlation of uncertainty between the two species that might cancel when calculating their ratio was neglected. Statistics of the resulting distributions in $BC/H_2O$ ratio after 50,000 iterations of the MC simulation are included in Table 2. Importantly, the experimentally measured variation in ratios for the realistic Bakken and

Ecuador flare gas mixtures is much larger than that which could be attributed to measurement uncertainty. The standard deviation was ~2.4–2.8 times larger and the skewness 4.6–7.2 times greater than what might be expected based on the Monte Carlo simulations (which represents a conservative upper bound on likely uncertainties as explained in Appendix A.5).



The measured BC/H$_2$O ratios of the Bakken and Ecuador flare gas mixtures, qualified by the control tests and Monte Carlo

simulations, point toward two important conclusions:

1)  Variability in instantaneous BC/H$_2$O ratios suggests the species are not always well-correlated in the turbulent plume; and

2)  Skewness of BC/H$_2$O ratio suggests the existence of short bursts of high black carbon production that are not well-correlated with H$_2$O.

The measured variability in path-averaged BC/H$_2$O ratios exceeds estimated measurement uncertainty and variability of

control tests and the variation is not symmetrical about the mean. The positive skewness of the BC/H$_2$O ratio suggests that infrequent bursts of high BC/H$_2$O ratio heavily weight the mean value and populate the long tail of the distributions observed in Fig. 8. This is interpreted physically as short bursts of high black carbon production that are not well-correlated with H$_2$O. Short, infrequent, and localized bursts of high BC production have been previously observed by Conrad and Johnson (2017) from sky-LOSA techniques measuring black carbon production from flares in Veracruz, Mexico and Orellana, Ecuador. Their

study demonstrated that overall BC emission rates are dominated by short bursts of high BC emission; in an extreme case, 10% of their instantaneous data corresponded to 56% of total emissions measured. What was not known in their study was whether these high-sooting bursts were necessarily correlated with spikes in concentration of other combustion species such that any ratio of species would be fixed. Experimental evidence from the current study suggests that this is unlikely. The observed variability and positive skewness in the distribution of the measured BC/H$_2$O ratios suggests that that assumption of

a constant species ratios is not universally true which could limit the accuracy of field measurement techniques quantifying flare performance based on short-duration measurements.

### 4.1 Implications for Field Measurements

The motivation of this study was to determine whether variations in species ratios in a turbulent plume could be detected, and if so, to examine how these might affect field measurements that rely on the assumption of a fixed ratio as part of a molar- or

mass-balance to quantify pollutant emission rates or combustion efficiency, etc. Although a sufficiently long temporally- and spatially-averaged species ratio measurement will approach some mean value for a given operating condition, the necessary measurement time- or length-scales required for accurate estimates are not clear. Moreover, the effects of fluctuations in relative species concentrations (*i.e.* species ratios) are exacerbated in measurement techniques that rely on a limited number of measurements. For example, airborne measurements to obtain extractive samples during plume transects (*e.g.* Gvakharia

et al., 2017; Weyant et al., 2016) are often limited by cost and airtime, deriving data from 1 to 6 plume transects per flare.

As a thought experiment, if we assume that an aircraft or similar path-averaged technique randomly samples 1 to 6 draws from the aggregated BC/H$_2$O histograms for the Bakken and Ecuador fuel mixtures (Fig. 8), this small sample size is unlikely to obtain an accurate estimate of the mean BC/H$_2$O ratio. The skewed distribution means the calculated black carbon emission

rate from a small number of samples would more likely be biased low, potentially missing infrequent bursts of high BC production. For any single measurement the most likely result would be the mode, which underestimates the mean ratio by



~13% for either fuel mixture. More importantly, however, the present results show how the broad variation in the ratios of instantaneous path-averaged species concentrations could introduce substantial additional uncertainty to measurements that assume these ratios are fixed.


A simple bootstrapping test was conducted to further consider the implications of undersampling. If we assume that some path-averaging measurement technique is sampling from either of the $BC/H_2O$ ratio distributions in Fig. 8, these distributions can be randomly sampled (with replacement) to derive statistical criteria for the minimum required samples size required for a target level of accuracy. Histograms of the relative error of the calculated mean $BC/H_2O$ ratio for a range of sample sizes

(each derived using 10,000 bootstrap trials) are plotted in Fig. 9. As should be expected, the potential error rapidly decreases as the sample size is increased. However, to be within a 10% error of the true mean $BC/H_2O$ ratio, the results of Fig. 9 suggest ~100 measurements would be required. For a measurement limited to 5 samples, relative errors between –34% and +35% could be expected (95% confidence limits of the distribution), with the most likely outcome (modes of Fig. 9) being a 1–5% underestimate of the true value. With only two samples, this range of likely errors grows to –55% to +53%, with the modes

dropping to 2.5–7.5% below the mean.

It should be reiterated that these results are based on path-averaged measurements through vertical flare plumes. Weak instantaneous species correlations are arguably of even greater concern for techniques that employ point-sampling (*e.g.* drone-based measurements, Krause and Leirvik, 2018), which do not benefit from long path averages to smooth out relative species

variations. Indeed, time-averaged measurements of the compositional structure of flare plumes in a wind-tunnel show that spatial variation in relative species concentrations (specifically methane and $CO_2$) can be very large and, on their own (*i.e.* ignoring additional temporal effects), are sufficient to produce substantial variations in the locally calculated carbon conversion efficiency at different points within the plume (Poudenx, 2000). More generally, the influence of crosswind on the flare plume could be expected to exacerbate the presently observed temporal variations in relative species concentrations. While turbulent

plume motion could have beneficial mixing effects for some measurement approaches, this is not necessarily the case. High-frequency methane sampling measurements (Johnson et al., 2001) and flow visualization experiments (Johnson and Kostiuk, 2002) of flares in crossflow have also shown that methane can be emitted as intermittent bursts below the lee-side of a deflected flare flame, where these bursts are separated from the main $CO_2$-rich plume. These observations also suggest that the presently measured variation in $BC/H_2O$ ratios is likely to extend to other species, notably unburned fuel and $CO_2$. Above all, the present

results suggest a need to carefully consider convergence criteria when designing and deploying measurement techniques that rely on an implicit assumption about temporally and spatially constant ratios of plumes species and a need for caution when interpreting measurements based on limited numbers of short-duration samples.


## 5    Conclusions

Experiments to measure the correlation between species in plumes of vertical, turbulent, buoyancy-driven flares have revealed that instantaneous, path-averaged concentrations of black carbon (BC) and $H_2O$ can vary independently and are not necessarily well-correlated over short time intervals.  Measurements using a novel tunable diode laser based optical system found considerable scatter in the $BC/H_2O$ ratio along a path through flare plumes that was well beyond that which could be attributed to measurement uncertainty.  Moreover, the distributions of $BC/H_2O$ ratios suggest BC emissions are skewed by intermittent bursts or fluctuations of high intensity, consistent with previous field observations.  These results suggest that the common assumption of fixed species ratios employed in many field measurement techniques is not universally valid and measurements based on small numbers of samples, short time intervals, and/or limited spatial coverage of the plume would be subject to potentially large added uncertainties.  The positive skewness of the distribution of $BC/H_2O$ volume fraction ratios also means that techniques relying on limited numbers of samples are more likely to risk introducing a negative bias in inferred BC emission rates.  However, a bootstrap analysis of the present results also demonstrates how these issues can be readily avoided with sufficient sample size and provides initial guidance for developing protocols for future field experiments using similar path-averaged measurement approaches.  Further study of the time- and length-scales that influence independent species fluctuations are warranted, especially in the context of single-point sampling approaches (potentially including sampling via remotely piloted aerial systems) which would not benefit from the optical path-averaging employed in the present experiments.

## Appendix A    Quantification of Measurement Error

### A.1    Spectral Profile Fitting Simulations

The performance of the optical measurement approach was first proven using large eddy simulation (LES) datasets of methane-rich flares in crossflow described by Conrad et al. (2020).  These simulations were of a 10-m tall, 102.3-mm diameter vertical flare stack in crosswind and were designed to yield data representative of gas flaring operations in Alberta, Canada.  The simulated flare burned natural gas (90.53% methane, 3.73% ethane, 0.34% propane, 1.54% $CO_2$, and 3.86% $N_2$) at a stack exit velocity of 4 m/s.  The present study considered two of the crosswind cases, 1.11 m/s and 7.78 m/s, corresponding to the 10th and 95th percentile windspeeds (simulations I and IV) in Conrad et al. (2020).  Because the available simulation domain was larger than the flare plumes produced at the Carleton University Flare Facility, the spatial domain was scaled down by a factor of two (5 cm grid resolution was assigned to be 2.5 cm) noting that this would only increase the intensity of the variation along the line-of-sight when testing the performance of the analysis algorithm.

Instantaneous line-of-sight (LOS) datasets from many different chords through the simulated flare plume were used to obtain a wide variety of test conditions.  These included samples in/near the flare where peak temperatures were high and combustion species were localized, as well as far-field samples where gas mixing had drastically reduced peak temperatures and



combustion species were more dispersed along the LOS. The extracted LOS datasets from the flare simulations were 150 cm

long, the intended path length of the experimental setup, and were 60 nodes across. Theoretically measured $H_2O$ spectra (*i.e.* the theoretical spectra reaching a detector after passing through the simulated plume domain) were generated by taking the product of the transmissivity through each node on the LOS, computed using the $H_2O$ volume fraction and temperature and assuming a Lorentz lineshape. Because the flare simulations did not include detailed aerosol modelling, the spatial distributions of black carbon volume fraction were produced by scaling the combustion-derived $CO_2$ volume fraction by a

factor of 1/750,000 [−] to produce the optical attenuation values between 0 and 5% expected in laboratory experiments, similar to Conrad et al. (2020). It was not important that the absolute BC volume fractions be accurate, only that the distribution be representative of what might be found in a flare to facilitate testing of the analysis techniques. Black carbon attenuation along a LOS was calculated as the product of the BC attenuation at each node, knowing the local BC volume fraction, and assuming BC optical properties from the literature. BC extinction coefficients, $K_{e_\lambda}$, were assumed for each measurement wavelength

using complex index of refraction values calculated from Chang and Charalampopoulos (1990), and a BC scattering-to-absorption ratio estimated from Migliorini et al. (2011). The theoretical 1428-nm detector signal was a product of both BC attenuation and $H_2O$ spectral absorption, whereas the 1654-nm signal was affected by BC attenuation only. Each of the calculated attenuation signals were intentionally corrupted with Gaussian-distributed noise with a standard deviation of $6\times10^{-4}$ [−]. The level of added noise used was conservatively selected to be 50% larger than that measured during tests of the

experimental system. The ability of the analysis algorithm to correctly reproduce the path-averaged concentration from the noise-affected integrated signals was then quantitatively tested.

In the ~8,000 instantaneous simulation results, true path-averaged BC volume fraction results ranged from 0 to 15 ppb and were correctly estimated to within 0.02 ppb in all cases. The error in instantaneous path-averaged BC volume fraction estimates

was solely due to precision error caused by the corrupting noise. Path-averaged BC volume fraction estimates were then used to isolate the $H_2O$ spectral absorption signal, as discussed in Sect. 3.2. The isolated $H_2O$ signals were fitted with theoretical column distributions of $H_2O$ volume fraction and temperature using a Nelder-Mead optimization scheme; the final calculated path-average $H_2O$ fraction and temperature were then compared to the known path-averages from the LES simulations. Figure A1(a) shows a typical result of the column distribution fit, plotting the theoretical spectral absorption fitted to the simulated

measured signal. Figure A1(b) and (c) compare the fitted column distributions of $H_2O$ volume fraction and temperature with the spatially-distributed LES data used to generate the measured signal. As indicated in the legend of Fig. A1(b), the path-averaged volume fraction in this example was estimated to within 1.2% of the true path-average. Similarly, the effective mean temperature was within 2 K of the true path-averaged value in Fig. A1(c).

By compiling all individual results, a trend of spectral fitting performance with peak gas temperature along the LOS for each measurement was observed. Figure A2 shows the relative error in path-averaged $H_2O$ as a function of the peak gas temperature. The error in $H_2O$ volume fraction appears to increase significantly with peak temperatures above roughly 800 K, which occurs



in approximately 5% of the simulation results. This is likely due to the divergence of the transition linestrengths from near-linear beyond the range of 300 to 800 K. In practice, 95% of simulated measurements with peak LOS temperatures above

800 K produced a path-averaged temperature above 315 K, the median path-averaged temperature was 351 K. By comparison, 95% of the experimental path-averaged temperatures were below 334 and 333 K for Bakken and Ecuador tests, respectively. This gives some indication that peak temperatures in the experiment were not in excess of 800 K although these results are difficult to compare directly. However, as previously discussed, thermocouple measurements in plumes downstream of turbulent jet diffusion flames by Poudenx et al. (2004) suggest that temperatures do not typically exceed 600 K. Based on

these results, the peak temperatures in the measured region of the present plumes are not expected exceed 800 K. Removing spectral fitting results from in-flame and with peak LOS temperatures in excess of 800 K, 95% of the ~7500 path-averaged $H_2O$ volume fraction results were estimated to within ±3.2% with mean relative error of 0.32%; the maximum relative error was 6.3%.

An additional potential source of measurement error could arise due to temporal changes in the individual spectral signatures within the 5 ms (200 Hz) time-resolution of the $H_2O$ measurement. Because the available LES data did not have sufficient time-resolution to investigate this issue, experimentally measured time-resolved BC attenuation data was instead used to produce synthetic time-varying $H_2O$ volume fraction data to estimate effects of temporal averaging of spectral signals. As discussed further in Seymour (2019), the results suggest a mean relative error of 0.28% for path-averaged $H_2O$ volume fractions

where 95% of measurements were within ±5.1%. Assuming temporal and spatial error estimated for path-averaged $H_2O$ volume fraction are independent, the values were combined in quadrature that resulted in a mean error of 0.63% with 95% of measurements falling within ±8.1%.

## A.2    Beam Steering Effects on Measurements

Although the optical system was meticulously aligned prior to each experiment and simulations by Conrad et al. (2020) suggest

beam steering through turbulent flare plumes should not exceed 1 mrad, beam steering effects in the present experiments cannot be completely discounted because the location and orientation of the highest temperature gradients is invariably unknown. Because beam steering would result in a loss or gain of light into the detectors, it would necessarily be interpreted as fluctuations in plume transmissivity and therefore BC presence. To experimentally quantify light loss due to beam steering, tests were completed on a flare burning pure hydrogen at 150 SLPM on the 25.4-mm diameter burner. The 1654-nm laser was

then used to acquire transmissivity measurements in the absence of BC. A histogram of laser signal attenuation, and apparent BC presence, for a 2-minute period is shown in Fig. A3. In this hydrogen flare test, where no BC is produced, the system observes an apparent path-averaged mean BC volume fraction of $-2.1 \times 10^{-12}$ (*i.e.* centered about zero) with a standard deviation of $3.2 \times 10^{-10}$. By comparison, the Bakken and Ecuador tests showed variations of $1.03 \times 10^{-9}$ and $1.66 \times 10^{-9}$ centered about mean values of $2.39 \times 10^{-9}$ and $3.69 \times 10^{-9}$, respectively. Therefore, 95% of the measurements presented





in Sect. 4 for Bakken and Ecuador flare gas have a BC signal-to-noise ratio (SNR) of at least 4 times the $H_2$ standard deviation; the median SNR is ~12.

Although the two laser beams were collinear through the heated plume, they were ultimately focused onto two separate targets meaning that it was still possible to have some relative differences in the signals due to beam steering. To bound the potential
impact of this effect, attenuation differences between the two laser signals were recorded through plumes of a hydrogen flare, whose adiabatic flame temperature (2379 K) exceeded that of all other flare tests, providing an upper limit to the magnitudes of temperature gradient-driven beam steering. The 1428-nm laser was tuned to a minimum $H_2O$ absorption wavelength of 7009.25 cm$^{-1}$ (1426.69 nm), where it was estimated that the $H_2O$ fluctuations would not contribute absorption magnitudes higher than 0.03%, and attenuation data was recorded for both laser lines. The differences in attenuation magnitude between
synchronous points from each laser were centred about zero (mean of 0.007%) with a standard deviation of 0.16% suggesting that 95% of measurements had attenuation differences of less than 0.31%. For reference, the main peak of interest to be measured varied between 2 and 6% absorption, *i.e.*, between 6.5 and 20 times higher signal amplitude in all cases. Although this will cause some distortion in the $H_2O$ absorption spectra, the profile fitting algorithm with goodness-of-fit filter was able to identify cases where beam steering produced poor fitting results as explained in Section A.4 below.

**A.3      Black Carbon Optical Property Ratio Measurements**

As discussed in Section 3.2, a ratio of black carbon (BC) extinction coefficient ($K_{e_\lambda}$) values at the two laser wavelengths was required to isolate the $H_2O$ absorption spectra in the presence of broadband BC attenuation. Since there was disagreement in the literature about BC optical properties in the near infrared, the ratio of optical properties used in the present analysis was measured experimentally. Using a heavily-sooting ethylene flare at 50 SLPM on the 76.2-mm diameter burner attenuation
measurements were recorded at the two laser wavelengths, 1654 nm and 1428 nm. The 1428-nm laser was tuned off of detectable $H_2O$ absorption peaks to solely measure BC attenuation through the plume. The extinction coefficient ratio (ECR) between the measurement wavelengths was calculated for conditions with at least 5% attenuation along the path (giving a SNR of >30) using Eq (1) and the experimental data are shown in Fig. A4 overlaid with values calculated from the literature. The mean BC extinction coefficient ratio ($K_{e_{1428}}/K_{e_{1654}}$) was 1.12 with a standard error of the mean of $1.3\times10^{-4}$. The resulting
ECR estimate is bounded by values calculated from extinction experiments in the literature (Table A1) and most closely resembles the result calculated using data from Krishnan et al. (2001). This necessary result permits the isolation of spectral $H_2O$ absorption in subsequent measurements.

**A.4      BC/H₂O Ratio Data Filtering**

As noted in Sect. 3.4, experimental data were filtered based on the closeness of fit between the measured $H_2O$ spectra and
theoretical spectra produced by the Nelder-Mead fitting algorithm, as characterized by the scalar sum-of-squared errors (SSE).



The purpose of this filtering was to reduce uncertainty by removing cases where the spatial or temporal changes in $H_2O$ volume fraction and/or temperature rendered spectra difficult to describe with a column distribution. As argued in the main text, a conservative SSE cut-off of $10^{-4}$ [-] was applied, following an investigation of the effects of the chosen SSE cut-off on distributions of the measured $BC/H_2O$ ratio.


The maximum SSE cut-off of $10^{-4}$ [-] was selected, corresponding to the largest SSE value measured during controlled $H_2O$ vapour measurements in an optical flow-through cell (Seymour, 2019). This implied an expected upper bound error for fitting of real-world spectra with theory. Subsequent analysis revealed that this choice is conservative, such that raising this filter threshold only increased the range of measured $BC/H_2O$ ratios and heightened the skewness of the distribution. This was

demonstrated by reprocessing the experimental datasets with a range of SSE filter values and recomputing the $BC/H_2O$ ratio variability and skewness. Figure A5 shows the filtering effect on standard deviation (a-b) and skewness (c-d) for the two fuel compositions. Less restrictive SSE filter settings only increase the variability and skewness of the $BC/H_2O$ ratio data.

### A.5 Upper Bound Potential Influence of Uncertainty on $BC/H_2O$ Ratio Distributions

To bound the potential influence of measurement uncertainty on the measured variations in the $BC/H_2O$ ratio, a simple Monte

Carlo simulation was conducted. As described in Sect. A.1, the $H_2O$ measurement had a mean relative error of 0.63% with a standard deviation of 4.1%; for BC, the mean relative error was 0.007% and the standard deviation was 0.16%. In Sect. A.2, hydrogen flare experiments suggest beam steering produces an apparent path-averaged mean BC volume fraction of no more than $-2.1\times10^{-12}$ with a standard deviation of $3.18\times10^{-10}$. Random draws of these experimentally-determined measurement uncertainties were independently applied to the median BC and $H_2O$ volume fraction results from the Bakken and Ecuador

datasets. The distribution of $BC/H_2O$ ratios about the median $BC/H_2O$ ratio (*i.e.* the distribution of results attributable to measurement uncertainty alone) were then plotted and overlaid on the experimental results for the Bakken and Ecuador fuel mixtures as shown in Fig. A6. It should be noted that by assuming measurement uncertainties on the BC and $H_2O$ signals are completely uncorrelated, this approach gives a conservative upper-bound on the potential effect of uncertainty on the $BC/H_2O$ ratio measurement.


As summarized in Table 2 of the main text, the Monte Carlo results, suggest that the maximum variation in the measured $BC/H_2O$ ratio that could be attributable to uncertainty had a standard deviation of $2.3\times10^{-7}$ and $3.1\times10^{-7}$ and skewness of 0.35 and 0.50 for Bakken and Ecuador mixture experiments, respectively. The Bakken and Ecuador experiments exceeded the Monte Carlo standard deviations by factors of 2.4 and 2.8, and skewness by factors of 7.2 and 4.6, respectively. As overlaid

in Fig. A6, the experimentally observed variation in the $BC/H_2O$ ratio is notably larger, and the skewness notably stronger, than the upper bound estimate of what might be attributable to measurement uncertainty alone.

*Acknowledgements:*



We are indebted to Prof. Jeremy Thornock and Prof. Philip Smith (University of Utah) for sharing high-fidelity LES data used
during test simulations to develop the optical diagnostic.  We are also grateful for the support and leadership of Michael Layer
(project manager, Natural Resources Canada) for championing this and related projects.


*Author contributions:*

MRJ and SPS contributed to the conception and design. SPS performed the acquisition of data. SPS and MRJ contributed to
analysis and interpretation of data, drafting and revising of the article, and approved the submitted version for publication.

*Competing interests:*

The authors have no known competing financial interests or personal relationships that could have appeared to influence the
work reported in this paper.

*Financial Support:*

This work was supported by the Natural Sciences and Engineering Research Council of Canada (NSERC) FlareNet Strategic
Network (Grant #479641), NSERC Discovery Research and Accelerator Supplement Grants (Grant #06632 and 522658), and
Natural Resources Canada (Project Manager, Michael Layer).

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





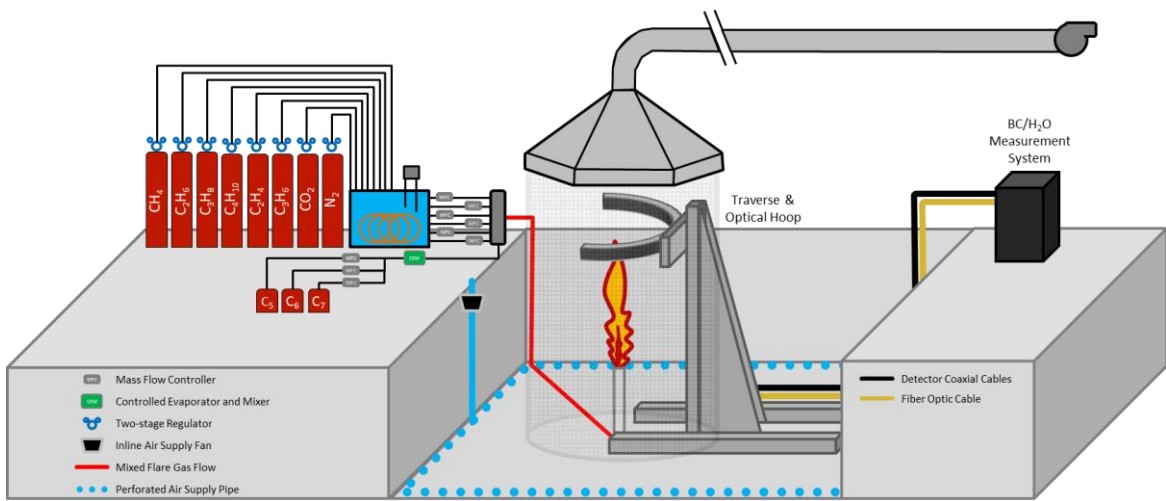


**Figure 1: Schematic of the Carleton University Flare Facility (CUFF) showing the mounting "hoop" for the optics that were positioned above the flare by a 3-axis traverse system.**

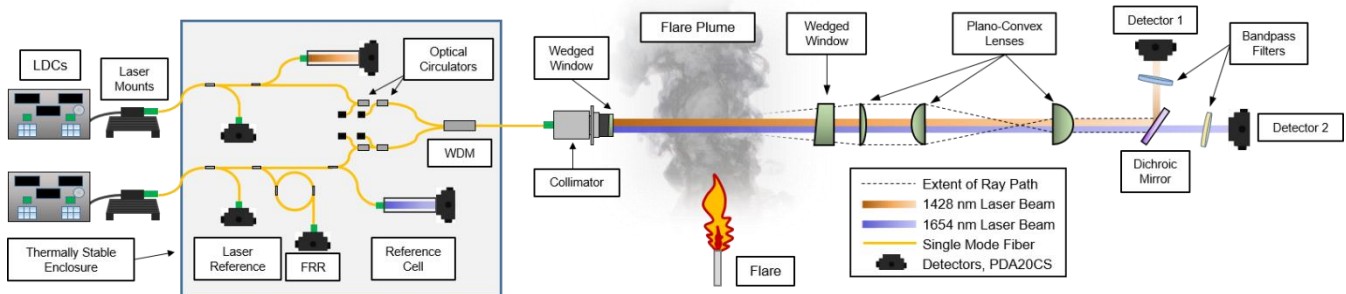

**Figure 2: Schematic of the optical measurement system for measuring synchronous path-averaged BC and H₂O volume fractions along a path through the flare plume. Components within the shaded box were temperature stabilized within a thermo-electric heater/cooler. Dashed lines representing ray path extent shown how optical arrangement minimizes any effects of beam steering through the plume.**



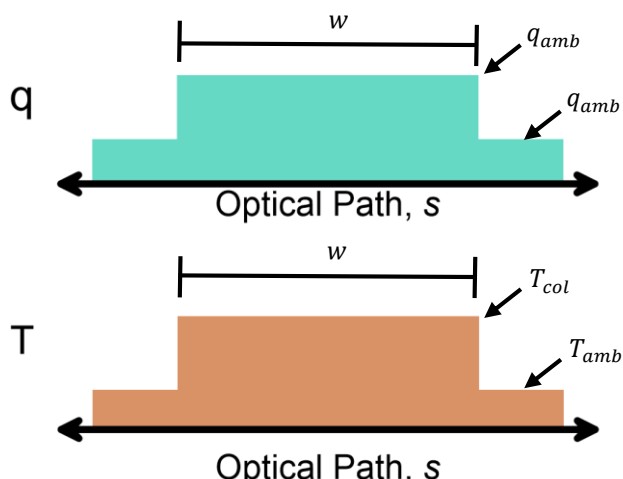

**Figure 3: Column distribution of temperature and $H_2O$ volume fraction used to generate theoretical spectra for comparison with measured data.**

**Table 1: Experimental fuel compositions in volume fraction [%] with their molecular weight (MW), volumetric higher heating value (HHV$_v$), and molar carbon-hydrogen ratio (CHR). Prefixes "n" and "i" distinguish between straight- and branched-chain isomers,**
**where appropriate, of the C1-C7 alkanes.**

| Fuel Name | C1 | C2 | C3 | nC4 | iC5 | nC6 | nC7 | H$_2$ | N$_2$ | CO$_2$ | CHR [–] | MW [g/mol] | HHV$_v$ [MJ/m³] |
|---|---|---|---|---|---|---|---|---|---|---|---|---|---|
| Bakken (BK)[a] | 49.12 | 20.97 | 15.04 | 6.72 | 2.15 | 0.87 | 0.78 | 0 | 3.66 | 0.70 | 0.331 | 29.13 | 64.33 |
| Ecuador (EC)[b] | 40.82 | 8.16 | 16.93 | 14.75 | 6.81 | 2.06 | 1.38 | 0 | 3.37 | 5.71 | 0.364 | 36.57 | 75.17 |

[a] Mean gas compositions from roughly 700 wells in Bakken region (Brandt et al., 2016)
[b] Selected gas composition from Ecuador's Orellana province (Conrad and Johnson, 2017)





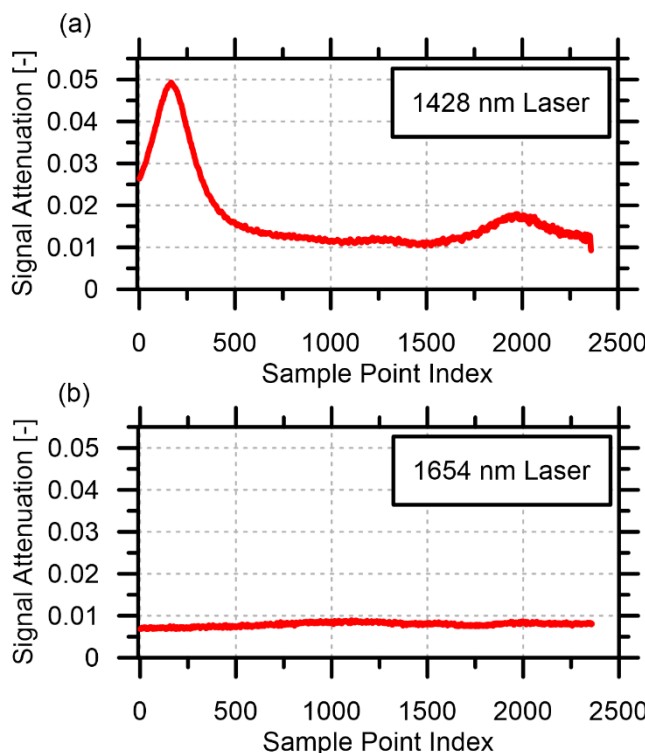

**Figure 4: Sample 5 ms duration measurement of (a) 1428 nm (H₂O+BC) and (b) 1654 nm (BC) laser signals through a Bakken flare plume. Time-resolved signals can be mapped to isolate spectral H₂O absorption peaks from black carbon.**

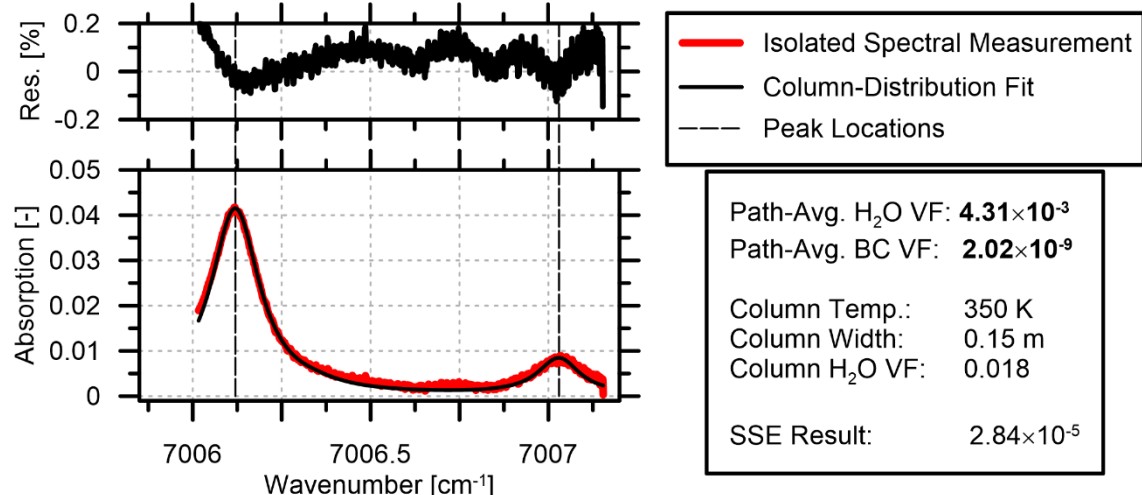

**Figure 5: H₂O absorption peaks in isolated, spectrally-resolved form compared to the column distribution result from the NM optimization search.**





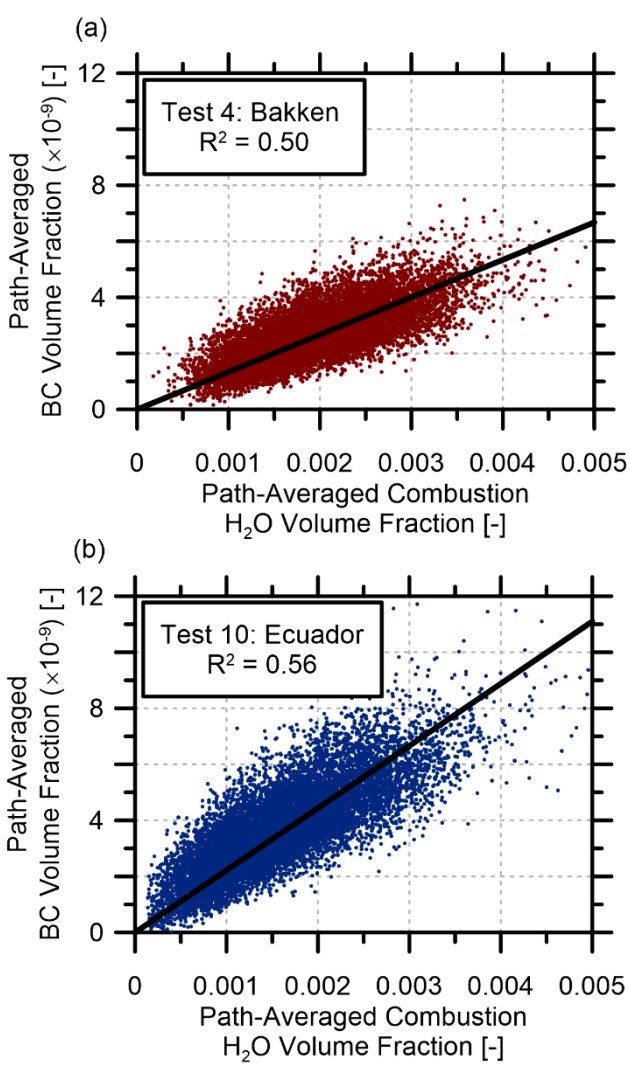


**Figure 6: Sample scatter plots of path-averaged produced black carbon (BC) and H₂O volume fractions for (a) Bakken fuel test 4 (40 SLPM, 25.4 mm burner), (b) Ecuador fuel test 10 (40 SLPM, 76.2-mm burner).**

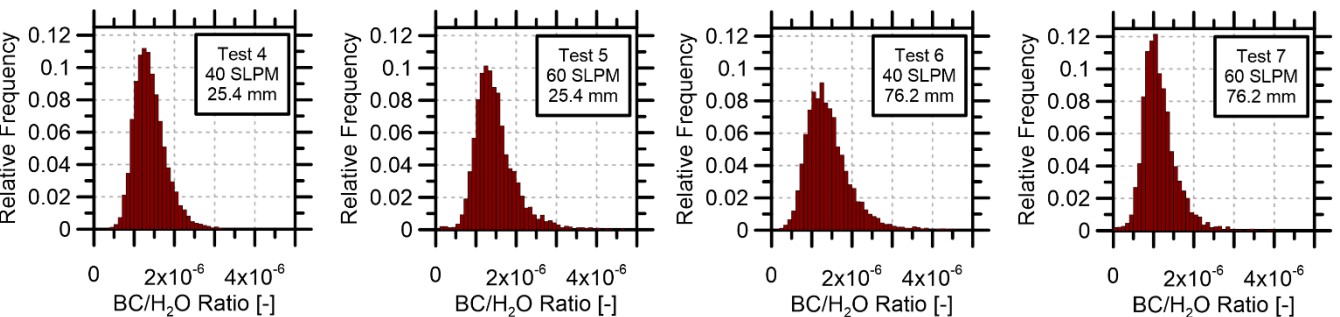



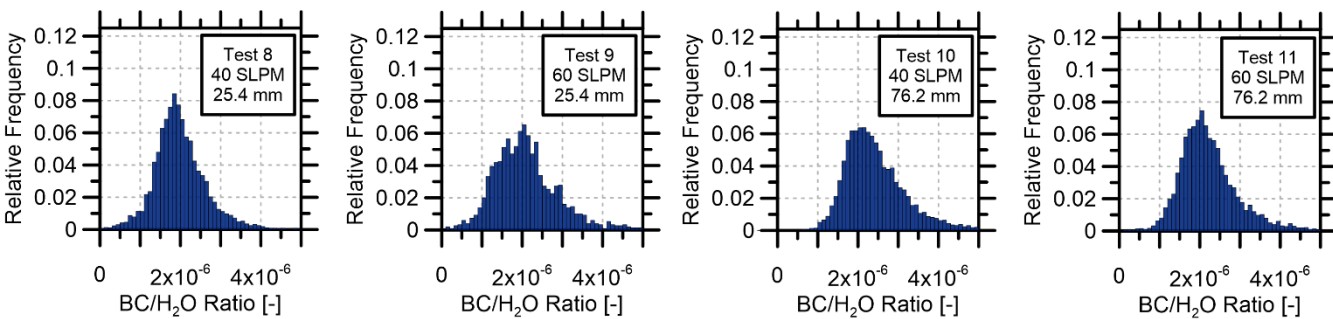


**Figure 7: Black Carbon (BC)/H₂O volume fraction ratios for tests with the Bakken flare gas mixture (top row), and Ecuador flare gas mixture (bottom row).**



**Table 2: Summary of Flare Test Black Carbon/H₂O Ratio Results and Relevant Statistics**

| Test ID. | Fuel Type | Test Condition | Points | Mean BC/H₂O Ratio [-] ($\times 10^{-7}$) | BC/H₂O Std. Dev. [-] ($\times 10^{-7}$) | Coeff. of Variation [-] | Skewness |
|---|---|---|---|---|---|---|---|
| Control-1 | Hydrogen | 150 SLPM, 25.4 mm | 9613 | 0.11 | 1.13 | 10.27 [b] | 1.47 [b] |
| Control-2 | Methane | 40 SLPM, 25.4 mm | 2349 | 1.11 | 0.85 | 0.77 [b] | 1.23 [b] |
| Control-3 | | 60 SLPM, 25.4 mm | 7426 | -0.14 [a] | 0.77 | -5.50 [b] | 1.06 [b] |
| Exp-4 | Bakken | 40 SLPM, 25.4 mm | 13015 | 14.06 | 4.17 | 0.30 | 1.53 |
| Exp-5 | | 60 SLPM, 25.4 mm | 4019 | 14.95 | 6.46 | 0.43 | 3.13 |
| Exp-6 | | 40 SLPM, 76.2 mm | 13656 | 14.34 | 6.31 | 0.44 | 2.36 |
| Exp-7 | | 60 SLPM, 76.2 mm | 3669 | 11.61 | 4.92 | 0.42 | 3.36 |
| Exp-8 | Ecuador | 40 SLPM, 25.4 mm | 4854 | 19.97 | 6.95 | 0.35 | 1.18 |
| Exp-9 | | 60 SLPM, 25.4 mm | 1526 | 21.26 | 9.34 | 0.44 | 1.89 |
| Exp-10 | | 40 SLPM, 76.2 mm | 14007 | 24.80 | 9.02 | 0.36 | 2.35 |
| Exp-11 | | 60 SLPM, 76.2 mm | 4747 | 22.72 | 8.93 | 0.39 | 3.10 |
| CH₄-Agg | Methane - Aggregated | Various | 9,775 | 0.16 | 0.95 | 5.94 | 0.96 |
| BK-Agg | Bakken - Aggregated | | 34,359 | 14.01 | 5.54 | 0.40 | 2.53 |
| EC-Agg | Ecuador - Aggregated | | 25,134 | 23.26 | 8.87 | 0.38 | 2.30 |
| BK-MC | Bakken – Monte Carlo | Various | 50,000 | 13.27 | 2.28 | 0.17 | 0.35 |
| EC-MC | Ecuador – Monte Carlo | Various | 50,000 | 22.46 | 3.13 | 0.14 | 0.50 |

[a] This non-physical value of the BC/H₂O ratio is a result of system noise and uncertainty on a BC signal near zero for the pure methane flame suggesting BC may not be sufficiently detectable for the methane experiment.

[b] Coefficients of variation and skewness for the control tests are included for completion but are not meaningful since the mean BC/H₂O ratios are near zero for these tests and the absolute variations (as measured by the standard deviation) are much smaller than those of the experimental cases.


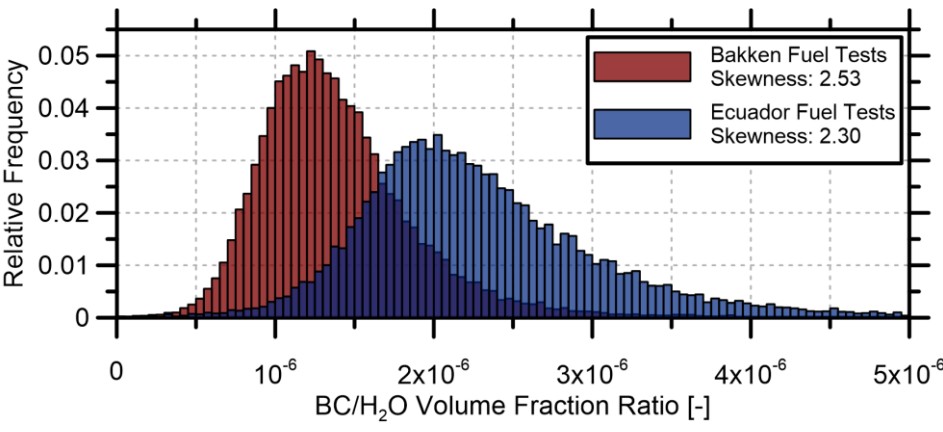

**Figure 8: Combined BC/H₂O ratio distributions for the two fuel types presented.**



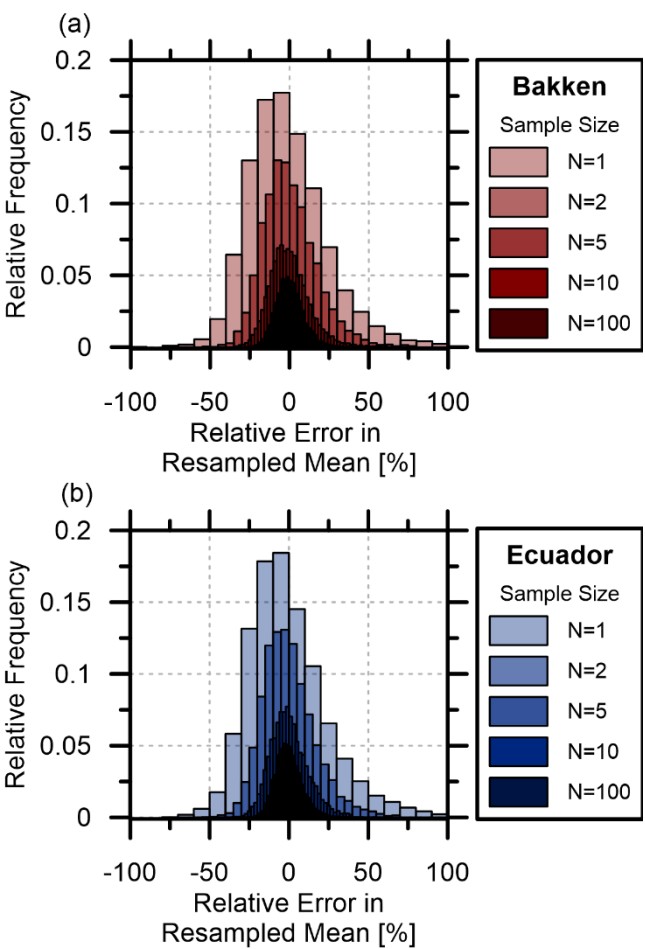


**Figure 9: Distributions of sample mean for sample sets of size *N* from BC/H$_2$O ratio results of (a) Bakken and (b) Ecuador flare tests.**





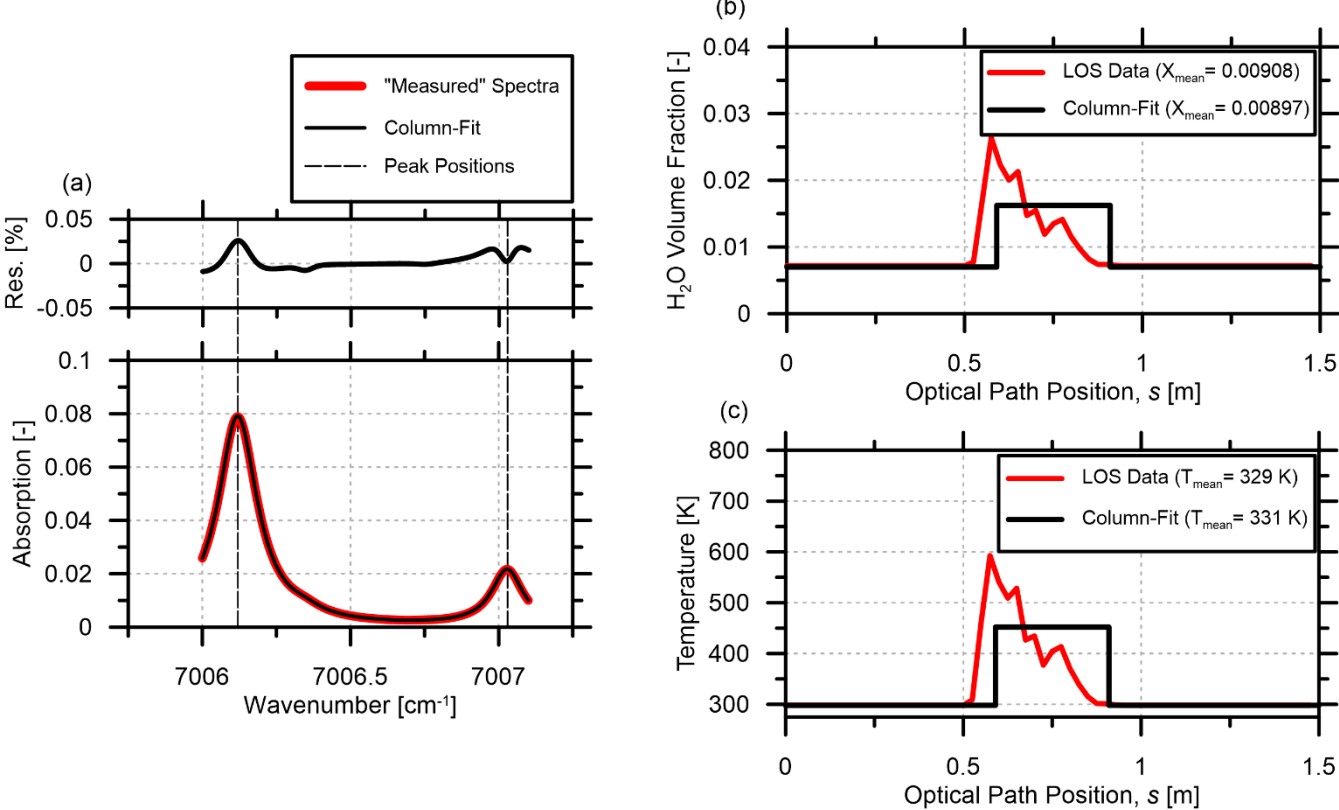


**Figure A1: Sample profile fitting result where (a) compares the column distribution spectra to the simulated measured spectra, and (b) and (c) compare the column distribution profiles with LOS simulation data for volume fraction and temperature, respectively.**

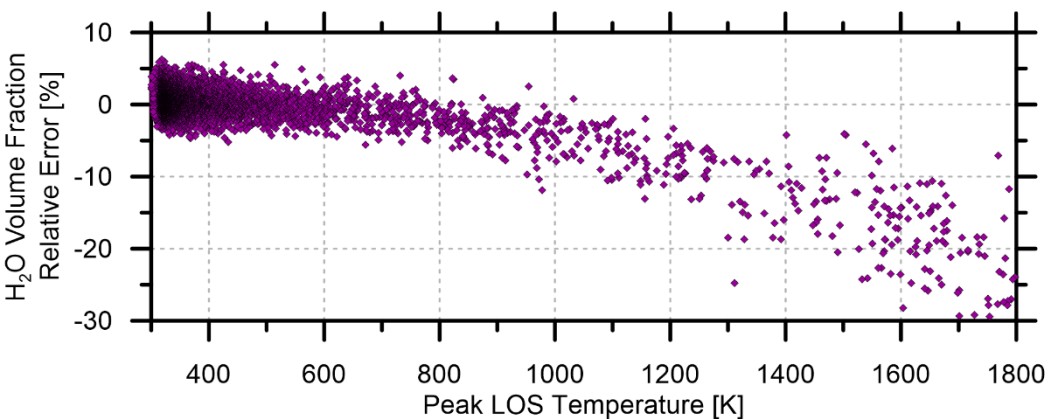

**Figure A2: Relative error of H₂O volume fraction results from simulated measured data compared to peak LOS temperature for each result.**


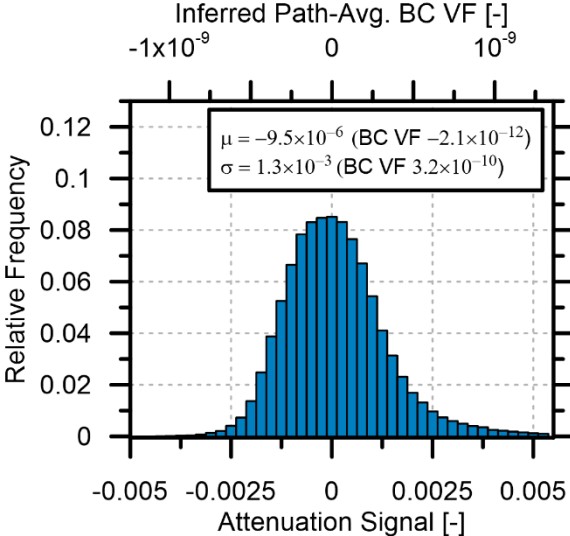

**Figure A3: Light attenuation due to beam steering and inferred black carbon (BC) volume fraction (VF) presence due to the attenuation.**


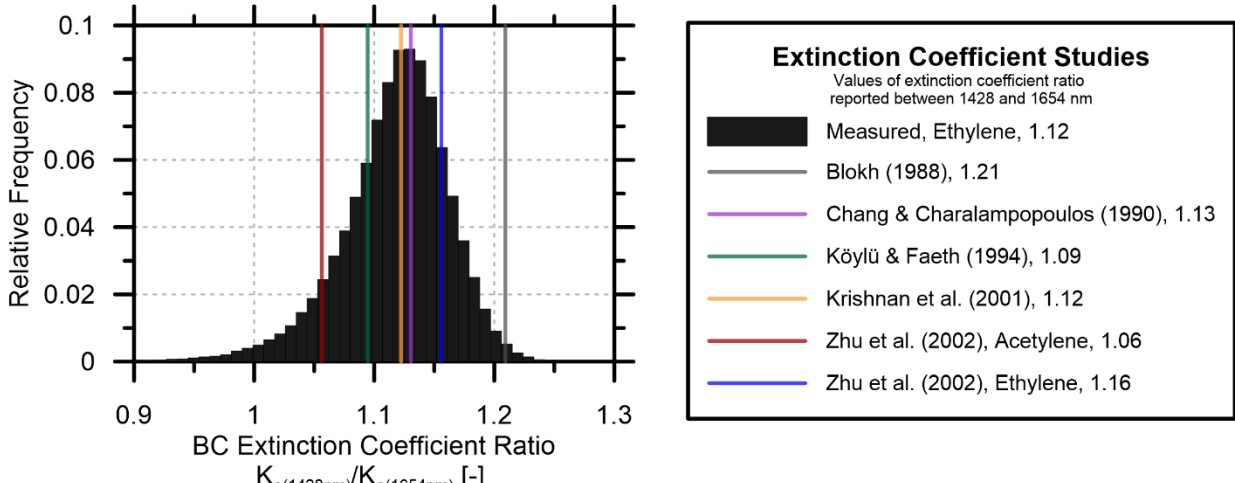

**Figure A4: Distribution of estimated $K_e$ ratio between 1428 and 1654 nm during ethylene flare test. The measured mean BC extinction coefficient ratio was 1.12 (standard error of the mean of $1.3×10^{-4}$).**





**Table A1: Summary of black carbon extinction coefficient measurements with extinction coefficient ratios (ECRs) calculated for the wavelengths selected for measurement.**

| Study | Method | Fuel Source | Wavelength Range, λ [μm] | ECR [-], $K_{e_{1428}}/K_{e_{1654}}$ |
|---|---|---|---|---|
| Blokh (1988) | Scatt/Extinction | Various | 1-6 | 1.21[a] |
| Chang & Charalampopoulos (1990) | Scatt/Extinction | $C_3H_6/O_2$ | 0.4-30 | 1.13 |
| Köylü & Faeth (1994) | Extinction | Various | 0.2-5.2 | 1.09[b] |
| Krishnan et al., (2001) | Scatt/Extinction | Various | 0.25-5.2 | 1.12[b] |
| Zhu et al. (2002) | Extinction | $C_2H_2, C_2H_4$ | 1.31, 1.56 | 1.06, 1.16[c] |
| **Present Study** | **Extinction** | **$C_2H_4$** | **1.428,1.654** | **1.12±1.3×10$^{-4}$** |

[a] Determined from complex index of refraction relations, assuming BC scatter-to-absorption ratio of 0.02.

[b] Calculated from interpolation of extinction data.

[c] BC ECR between 1565nm & 1314 nm.



**Figure A5: SSE-filtering effect on BC/H₂O ratio statistics. (a) and (b) show standard deviation for Bakken and Ecuador fuels, and (c) and (d) indicate skewness of ratio.**





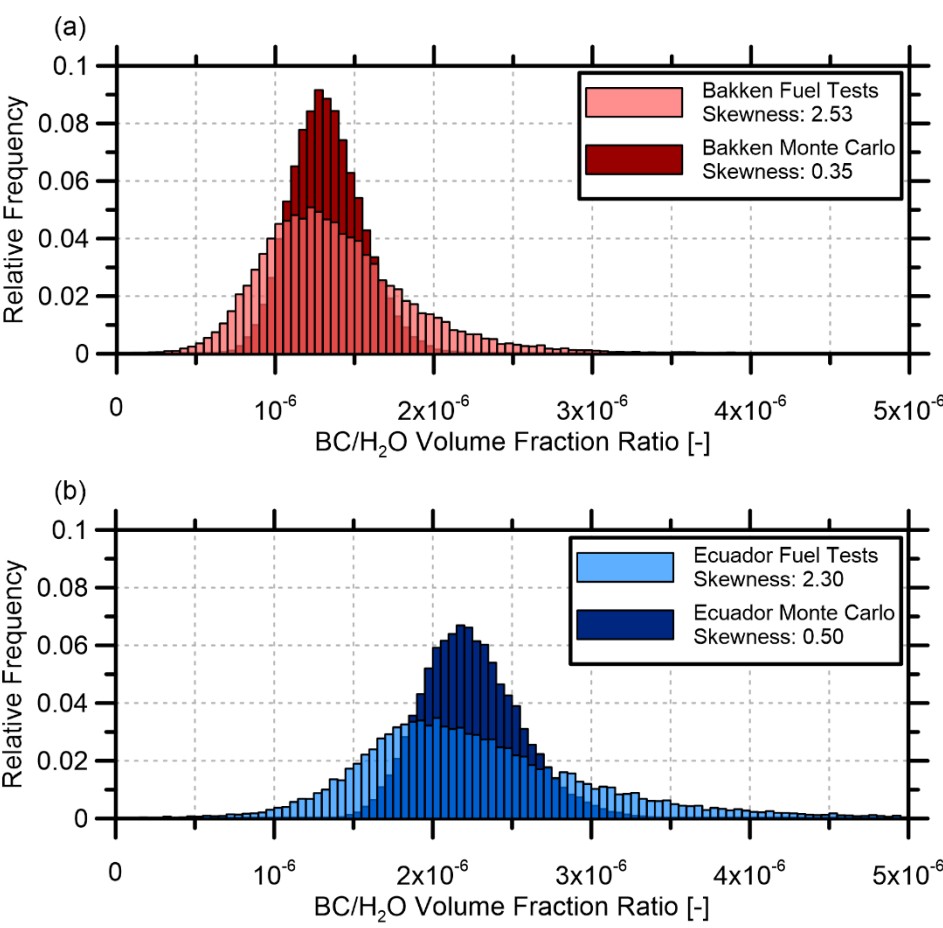

**Figure A6: Distributions of black carbon (BC)/H$_2$O volume fraction ratios from Monte Carlo simulations with error related to beam steering, system noise, and spatial- and temporal-averaging.**