# Peer review of "Species Correlation Measurements in Turbulent Flare Plumes: Considerations for Field Measurements"

_Atmospheric Measurement Techniques, 2020_

## Referee Comment (RC1)

**Review for Species Correlation Measurements in Turbulent Flare Plumes: Considerations for Field Measurements**

**General comments**

Field measurements of flaring emissions are challenging and unlike those faced by other pollution sources. There are only a few measurements in the literature and most use novel measurement techniques that are difficult to verify. This paper explores an understudied question of the emission variability and its impacts on current measurement techniques and assumptions. A novel measurement method was devised to determine the temporal variability in $BC/H_2O$ ratios measured simultaneously through a laboratory flare plume. The authors clearly present the measurement methods and thoroughly investigate method uncertainties. However, the conclusions and implications are weakly supported. In particular, the variability observed is applied to other sampling conditions without consideration of the effect that sampling technique has on variability.

**Specific comments**

1) The authors make the assertion that the variably observed in $BC/H_2O$ ratios suggests that fly-by sampling of flares (Gvakharia2017, Weyant2016, and Krause2018) may be prone to large uncertainties and low bias. While these flight passes are only a few seconds and, by most measures are very short samples, they are also much longer than the near-instantaneous optical measures used here. Additionally, the volume of the plume represented by the laser beam is tiny compared to that captured in the flight passes. Considering both the differences in duration and volumes, these flights may capture over 1000 times more of the plume than a single instantaneous optical measure. Aggregating just a few seconds of the optical data would reduce the variability observed, substantially.

   While levels of aggregation that better represent flight-passes could be explored in this paper, it isn't clear what amount of aggregation would be appropriate for both the temporal and volume issues. It may be a fundamentally apples-to-oranges comparison to use the variability observed at such small scales to infer the variability of samples made with much greater volumes and duration.

2) The paper seems to be missing arguments about the relationship between the $BC/H_2O$ measured here and the ratios used in the molar/mass balance literature, i.e. BC/combusted carbon or $BC/CO_2$. Are $BC/H_2O$ ratios equally as variable as $BC/CO_2$ ratios? It would be natural to assume that the $H_2O$ and $CO_2$ produced are well correlated from the combustion equation, but the paper currently doesn't mention that this relationship is necessarily assumed.

   Another potential related area to explore would be to use the $H_2O$ to infer $CO_2$ and provide BC emission factor estimates using carbon balance. Presentation in this metric could both better articulate the relevance of the $BC/H_2O$ metric and present variability using metrics readers may be more familiar with. It may also build confidence in the novel measurements conducted here, if they could be grounded around previously measured values.

3) A novel measurement technique is introduced, but not calibrated or verified using other measurement tools. The argument appears to be that because ratios, not absolute magnitudes are presented, this validation is not necessary(?). However, if there are multiplicative errors (instead of additive) then the $BC/H_2O$ ratio variability would be affected.

4) The paper repeatedly suggests that stable $BC/H_2O$ ratios are assumed when using molar/mass balance methods, and suggests that readers should change their understanding of the nature of flare emissions from stability to one that is highly variable. However, most emissions researchers would expect that BC emissions are highly variable; this is observed in all but perhaps the most controlled combustion conditions for any fuel. If stability was the expectation, researchers would be reporting the results of a single measurement, but they do not. (Maybe 1-6 passes of a single flare, but reporting regional averages). If the majority of

readers in the target audience do not have a prior belief that flare emissions are stable, then (in my opinion) it would not be a great choice to lean on this idea so heavily, as it may lose, or worse offend some readers.

**Technical corrections**

line 9: *"…emission rate (i.e. emission factors)…"* Change to "emission factors".

line 25: Acronym "UOG" not used again, so could be omitted.

line 28-29: 20/100 and 96/34 look like fractions. Perhaps another phrasing to avoid this potential confusion.

line 33: *"…of soot"*. Suggest changing to "of particulate matter", since sometimes soot is considered equivalent to BC.

line 34: Suggest "second most warming atmospheric pollutant", because there are other strongly negative climate forcers, and "important" is subjective.

line 46: Nearly all emission sources are "infrequently" measured; most individual cars, planes, industrial stacks are not measures or are infrequently measured. A stronger case may be made by quantifying the frequency of measurements relative to other sources.

line 61: I suggest changing the sentence *"The veracity of this…"* to "Assessing the veracity of this assumption is necessary to determine representative sampling strategies" (e.g. sampling volumes, number of samples, and perhaps sample timing)"

lines 59 & 63: *"instantaneous measurements"*. The above papers are not reporting instantaneous metrics as is being stated here.

line 66: *"..there has not been study attempting…"*. Change to "…no study has attempted to…"

line 67: "the first"

line 94: What is the likelihood that some of the water would have started to condense at this height? Could this impact the measured $H_2O$?

lines 113-114: Could the "<1 mrad beam steering that is interpreted as additional BC" be quantified in terms of measured BC? Or possibly as a fraction of the average measurement? It is hard for a non-specialist in this technique to interpret the magnitude of this error.

lines 128: Was the BC also averaged over 5 ms?

line 155: BC or soot volume fraction is a niche term. Why not use mass concentration?

line 165: Using a back of the envelope calculation, it seems like this extinction coefficient is large. If the mass extinction cross section $(m^2g^{-1})$ is around 4 and the particle density is 1.7g $cm^{-3}$, this value would be around $7x10^6$ $m^2m^{-3}$ $(m^{-1})$. Perhaps a more explicit explanation of the methodology used would clear up the confusion.

lines 167-175: What fraction of the light attenuation at the 1428 wavelength was estimated to be due to BC for a typical measurement?

line 241: What does "similarly averaged" mean? Was there also a wavelength sweep at 1654nm?

Figure 4: "Sample point index" is not very meaningful. Could this be plotted with time on the x-axis instead? My understanding is that this is the time within the 5ms sweep.

line 250: How much of the data was omitted due to the SSE threshold?

lines 260-267: This paragraph feels like it belongs in an "experimental design" section of the methods, rather than in Results.

line 271: A general correlation between the species is not expected "*since the production of BC necessitates the production of $H_2O$*". Most $H_2O$ is produced through the generation of $CO_2$. A tiny amount of BC is produced relative to $H_2O$, thus the variably in this ratio is directly related to the variability in BC formation, which varies due to micro-physical changes in combustion conditions. Whereas, I suspect the variability observed in $H_2O$ is primarily due to dilution. It does not seem surprising, or particularly significant that such variability was observed.

line 276-277: *"..is not explained by a linear model..."* Why should it be? Why isn't a large variance expected? Are we sure these $R^2$ values should be characterized as "low"?

line 286: Is there a theoretical reason why the ratios and variability were different between the two flare gas mixtures? This result seemed interesting enough to warrant inclusion of a figure, but the implications and interpretation are under-explored.

line 292-293: *"...driven by turbulent dynamics..."* I suspect that this is the main driver of variability, but it isn't clear to me how the comparable skewness of the distributions generated this conclusion.

line 331: At first glance, the skewness observed here appears to be lower than found by Conrad & Johnson (2017), which would suggest that high BC events are somewhat correlated with higher rates of combustion. How do these measurements compare to those observations?

line 349: After *"ratio..."*, add "for that individual flare".

line 366: This description of "path-averaged" and "point-sampling" doesn't correctly describe the differences between the drone samples and the path-averaged optical measurements described here. 1) The drone measurements (as well as those conducted with airplanes) are sampled through the plume, collecting a volume of air for the time it takes to pass through the plume. The assertion that these samples capture less of the plume than a single optical measurement is not convincing; the volume captured in the optical path-averaged measure is tiny. 3) It is also likely that the plume, measured further downstream than here is much better mixed than from 2m from the flare, such that even small samples are likely to have less variability in ratios. 4) The description here also implies that these measures report a single measurement as if it was representative, but they do not. Reporting is made of averages of a number of passes. 5) The description of the path-average sampling seems to suggest that more of the plume is measured than with other sampling techniques, but the volume represented by a laser beam across the plume is small and also fails to capture most of the plume, which varies spatially as well as temporally.

line 375-379: Is methane leaking germane to the discussion about combustion pollutant ratios? The idea that measured variability in $BC/H_2O$ extends to other species should be more grounded in combustion theory. I suspect the $BC/H_2O$ ratios have little bearing on the release of unburnt fuel, but are fundamentally related to the relative emissions of $CO_2$.

line 415: Is the 150 cm the distance from one side of the optical hoop to the other (from collimator to wedged window in Figure 2)? If so, what are the nodes?

line 434: How can you know the BC was *"correctly estimated to within 0.02 ppb"* without external calibration? Is this actually a measure of measurement uncertainty?

Figure A1b: It isn't clear how features within the beam are determined. Were they measured or modeled?

---

## Author Comment (AC1)

**Manuscript ID:** amt-2020-472
**Title:** Species Correlation Measurements in Turbulent Flare Plumes: Considerations for Field Measurements
**Authors:** Seymour, S.P.; Johnson, Matthew R.

**Point-by-point Response to Comments by Referee #1**

**General comments**

*Field measurements of flaring emissions are challenging and unlike those faced by other pollution sources. There are only a few measurements in the literature and most use novel measurement techniques that are difficult to verify. This paper explores an understudied question of the emission variability and its impacts on current measurement techniques and assumptions. A novel measurement method was devised to determine the temporal variability in BC/H₂O ratios measured simultaneously through a laboratory flare plume. The authors clearly present the measurement methods and thoroughly investigate method uncertainties. However, the conclusions and implications are weakly supported. In particular, the variability observed is applied to other sampling conditions without consideration of the effect that sampling technique has on variability.*

Thank you for your feedback and thorough review of our manuscript. We especially appreciate your recognition of the understudied nature of this important problem, the novelty of the approach, and the care and effort we put into the measurement method and uncertainties. We have responded in detail to each of your comments below and thank you again for your help in improving the clarity of the manuscript and further strengthening the discussion which we hope will draw attention to this important issue when designing future measurement campaigns.

**Specific comments**

1) *The authors make the assertion that the variably observed in BC/H₂O ratios suggests that fly-by sampling of flares (Gvakharia2017, Weyant2016, and Krause2018) may be prone to large uncertainties and low bias. While these flight passes are only a few seconds and, by most measures are very short samples, they are also much longer than the near-instantaneous optical measures used here. Additionally, the volume of the plume represented by the laser beam is tiny compared to that captured in the flight passes. Considering both the differences in duration and volumes, these flights may capture over 1000 times more of the plume than a single instantaneous optical measure. Aggregating just a few seconds of the optical data would reduce the variability observed, substantially.*

*While levels of aggregation that better represent flight-passes could be explored in this paper, it isn't clear what amount of aggregation would be appropriate for both the temporal and volume issues. It may be a fundamentally apples-to-oranges comparison to use the variability observed at such small scales to infer the variability of samples made with much greater volumes and duration.*

The presented optical system was not designed to reproduce aircraft measurements, although it should be noted that both measurement techniques effectively obtain path-averaged estimates of combustion species concentrations in the plume – the aircraft drawing a sample with a probe as it transects the plume, and the present optical method which obtains an instantaneous path-average measurement of the species along the laser path. More importantly, our experimental results show that the path-averaged BC/H₂O ratio is highly variable with a skewed distribution in the near-field of a turbulent flare plume. This raises a key issue to be considered in measurement techniques that make explicit or implicit assumptions about the stability of species correlations in a turbulent combustion plume.

Absolutely, we agree that sufficient data aggregation should yield representative results from field measurement techniques, but it is equally important to know or have a means to estimate what might constitute "sufficient" data. This is indeed a main point of this paper. Figure 9 of the manuscript suggests that for the present system, more than 50 path-averaged measurements would be necessary to ensure the mean BC/H₂O ratio is captured within 10% at 95% confidence. In this sense, it certainly seems possible that some of the variability in same-flare emission factors observed by Weyant et al. (2016), Gvakharia et al. (2017), or Caulton et al. (2014) could at least partially stem from a limited set of measurements made from a skewed distribution. Whether a difference in absolute sample volume as noted by the

reviewer would matter is not clear, and would depend on how the relative sample volume and overall plume dimensions are assumed to scale as the plume moves downstream; we have added this to our discussion in Sect. 4.1 (see also our response at line 366). Thus, for any given field sampling technique, having identified this potential issue, the best option would be to conduct convergence tests for that particular technique. However, in the absence of such data and/or as part of the design phase of future measurement campaigns, we offer the present results to the research community as a starting point.

*2) The paper seems to be missing arguments about the relationship between the BC/H₂O measured here and the ratios used in the molar/mass balance literature, i.e. BC/combusted carbon or BC/CO₂. Are BC/H₂O ratios equally as variable as BC/CO₂ ratios? It would be natural to assume that the H₂O and CO₂ produced are well correlated from the combustion equation, but the paper currently doesn't mention that this relationship is necessarily assumed.*

*Another potential related area to explore would be to use the H₂O to infer CO₂ and provide BC emission factor estimates using carbon balance. Presentation in this metric could both better articulate the relevance of the BC/H₂O metric and present variability using metrics readers may be more familiar with. It may also build confidence in the novel measurements conducted here, if they could be grounded around previously measured values.*

We agree with the reviewer and anticipate $CO_2$ is likely well-correlated with $H_2O$ in the plume, especially in comparison to the correlation of BC and $H_2O$ or BC and $CO_2$, which are expected to be highly variable. For any hydrocarbon fuel ($C_xH_y$), the primary combustion products and expected end states of the hydrogen and carbon atoms are $H_2O$ and $CO_2$. These are the species associated with the vast majority of the energy release in a flame (e.g. Turns, 2000) and are expected to be correlated and found in the same hot regions of the turbulent flame and plume. Indeed, the LES simulation data of flares in Conrad et al. (2018, 2020) show very strong correlation between $H_2O$ and $CO_2$ as they form in the flame ($r^2$ of 0.998), with the slight deviations being due to CO production. In the turbulent plume beyond the flame, $H_2O$ and $CO_2$ have an $r^2$ of 0.9998.

Given the expectation that $H_2O$ and $CO_2$ are almost perfectly correlated, the variability in the BC/$H_2O$ and BC/$CO_2$ ratios should be the same, and this variability is a direct indication of the variation in a measured BC emission factor based on either of these ratios. In other words, a 10% variation in the BC/$H_2O$ or BC/$CO_2$ ratio would be expected to induce a 10% variation in the estimated emission factor. We thank the reviewer for suggesting this clarification. We have added this discussion to our manuscript to more directly compare the BC/$H_2O$ variability to BC emission factor variability.

Regarding the question of converting BC/$H_2O$ ratios into an estimated BC emission factor, this is something we have attempted previously in Seymour (2019). The estimated emission factors were of the same order as those found by Conrad and Johnson (2017), however since their absolute magnitudes were dependent on assumed BC optical properties we felt these distracted from the present focus on species ratio variability.

*3) A novel measurement technique is introduced, but not calibrated or verified using other measurement tools. The argument appears to be that because ratios, not absolute magnitudes are presented, this validation is not necessary(?). However, if there are multiplicative errors (instead of additive) then the BC/H₂O ratio variability would be affected.*

The use of optical attenuation measurements to quantify gas and particulate phase species concentrations is a very mature and well-established approach (e.g. Grauer et al., 2018; Johnson et al., 2013; Rieker, 2009; Rieker et al., 2009; Zhang et al., 2016) and is the basis of a wide range of commercial instruments. While we did not calibrate our system in the simplest sense by directly comparing to some other measurement instrument (and note that there isn't an obvious choice for an existing instrument to use for such a comparison), it is not accurate to say that the system has not been verified or validated.

Critically, the use of light absorption/attenuation to infer the presence of species is a self-referencing measurement, where the measured *ratio* of the absorption-free intensity (measured at the beginning of each experiment as detailed in Sect. 2.2.) to the measured optical intensity during the experiment relates directly to the volume fraction of the absorbing species. The key advantage of this approach is that it does not require an absolute intensity calibration and ultimately relies on established spectroscopic theory and the linearity of the detector's response. For this latter point, silicon-based photodiodes like the ones used in the present system have long been relied upon to provide this critical linear response

to incident optical power. Specifically, the InGaAs photodiode type used in this study was demonstrated by Yoon et al. (2003) to have exceptional linearity (within 0.08%) in multiple photodiodes across four manufacturers. Moreover, the present experiment leveraged reverse-biased detectors which further improve the linearity across a larger incident power band (Quimbly, 2006). Even if the linear response of the two detectors were different, because of the self-referencing nature of the technique, the $BC/H_2O$ ratio would still be the same. Nevertheless, it is worth noting that the measured BC extinction coefficient ratio (see Appendix A.3) agrees closely with BC optical properties that can be calculated from the literature which adds further confidence to the approach.

Finally, and most importantly, as noted by the reviewer we have "thoroughly investigat(ed) method uncertainties". First, the employed techniques, which build directly off of theory and experiments by Wu et al. (2017) and the profile fitting techniques introduced by Liu et al. (2007), were meticulously verified using large eddy simulation data (Appendix A.1). Separate experiments were also conducted to bound beam steering effects on BC volume fraction estimates (Appendix A.2). Experimental results were filtered to ensure that presented estimates of species ratio variations were appropriately conservative (Appendix A.4). Finally, we conducted a detailed Monte Carlo simulation (Appendix A.5 and discussed in the main text) where sources of error were assumed independent which gave a *conservatively high* estimate of species ratio uncertainty which was used when discussing potential implications of the results.

*4) The paper repeatedly suggests that stable $BC/H_2O$ ratios are assumed when using molar/mass balance methods, and suggests that readers should change their understanding of the nature of flare emissions from stability to one that is highly variable. However, most emissions researchers would expect that BC emissions are highly variable; this is observed in all but perhaps the most controlled combustion conditions for any fuel. If stability was the expectation, researchers would be reporting the results of a single measurement, but they do not. (Maybe 1-6 passes of a single flare, but reporting regional averages). If the majority of readers in the target audience do not have a prior belief that are emissions are stable, then (in my opinion) it would not be a great choice to lean on this idea so heavily, as it may lose, or worse offend some readers.*

It is important to distinguish between stability of $BC/H_2O$ ratios and variability of BC emission rates which are potentially quite different. While time-resolved emission rate data such as those shown in Figure 2 of Conrad and Johnson (2017) show that instantaneous BC emission rates are indeed highly variable, to our knowledge there was little in the literature to determine the degree to which relative concentrations, i.e. $BC/H_2O$ or $BC/CO_2$, might vary. This distinction is important when interpreting field data.

For techniques relying on measured species ratios in a molar or mass balance, inherent variability in these ratios will manifest as an added source of uncertainty in the inferred emission rates. While some studies have explicitly acknowledged this potential source of error (e.g. Caulton et al., 2014; Pohl et al., 1986), this is not generally the case. Critically, for studies reporting results from limited samples or attributing variations among small numbers of passes to variations in emissions, there is an implicit assumption that species correlations are sufficiently strong that the available measurements will provide representative results. Echoing our response to Specific Comment 1, above, while we agree that regional averages are commonly the desired quantity to be reported, it is vital to understand the factors driving variability in the underlying data to give confidence in the final result. We appreciate the feedback of the reviewer and we have tried to make this more clear in the revised text. The essential point is that the observed species ratio variability is demonstrated as an important factor to be considered when designing field campaigns. This result underscores the importance of convergence tests to guard against potential biases and uncertainties from undersampling.

**Technical corrections**

*line 9: "...emission rate (i.e. emission factors)..." Change to "emission factors".*

Changed as suggested.

*line 25: Acronym "UOG" not used again, so could be omitted.*

Acronym UOG omitted as suggested.

*line 28-29: 20/100 and 96/34 look like fractions. Perhaps another phrasing to avoid this potential confusion.*

The values of global warming potential (GWP) of methane are presented in this way following the standardized notation set forth by Ocko et al. (2017). This standard reporting format was suggested to ensure that both time-horizon values were consistently and clearly reported together when discussing pollutant impacts.

*line 33: "...of soot". Suggest changing to "of particulate matter", since sometimes soot is considered equivalent to BC.*

Changed as suggested.

*line 34: Suggest "second most warming atmospheric pollutant", because there are other strongly negative climate forcers, and "important" is subjective.*

We have modified this line to read,

> *black carbon ... and methane are both Short-Lived Climate Pollutants (SLCP) that, along with $CO_2$, constitute the three most climate warming pollutants in the atmosphere (e.g. Bond et al., 2013; IPCC, 2013; Jacobson, 2010; Ramanathan and Carmichael, 2008).*

*line 46: Nearly all emission sources are "infrequently" measured; most individual cars, planes, industrial stacks are not measures or are infrequently measured. A stronger case may be made by quantifying the frequency of measurements relative to other sources.*

We have removed the cited sentence entirely, which is better stated and referenced in the final sentence of this paragraph:

> *Uncertainties in global pollutant emission inventories stemming from uncertainties in the few available flare measurements can have significant impacts on climate modeling studies (Klimont et al., 2017; Winiger et al., 2019).*

*line 61: I suggest changing the sentence "The veracity of this..." to "Assessing the veracity of this assumption is necessary to determine representative sampling strategies" (e.g. sampling volumes, number of samples, and perhaps sample timing)"*

We have amended this line to,

> *Assessing the veracity of this assumption is vital for designing robust sampling strategies to obtain representative flare performance metrics.*

*lines 59 & 63: "instantaneous measurements". The above papers are not reporting instantaneous metrics as is being stated here.*

Thank you for catching this. We changed "instantaneous measurements" to "a limited set of measurements" at line 63 and edited line 59 as follows,

> *Critically, these techniques all rely on the assumption that the species in these measurements are well-mixed, and therefore well-correlated, in the plume, or that sufficient measurements have been acquired to minimize the added bias and uncertainty from this assumption.*

*line 66: "..there has not been study attempting...". Change to "...no study has attempted to..."*

Typo corrected as suggested.

*line 67: "the first"*

Added missing "the" as suggested.

*line 94: What is the likelihood that some of the water would have started to condense at this height? Could this impact the measured $H_2O$?*

There is no chance of condensation. For the Bakken and Ecuador fuel compositions, the $H_2O$ volume fraction in the undiluted combustion products would be ≤16.6 and 15.7%, respectively (i.e. maximum possible values assuming complete combustion) with an initial adiabatic flame temperature of 1971–1974°C. Before allowing for dilution, $H_2O$ would only condense below 100°C at ambient pressure. For these flare gas mixtures, radiation heat transfer would be no more than

5–45% of the total heat release rate (e.g. Turns, 2000), such that the initially undiluted combustion products would have a temperature of at least 1060°C. Any further cooling of the plume occurs via mixing and dilution with lab air at ~20°C and <60% relative humidity (typically 30-50%). While the temperature will decrease as the combustion products mix with ambient air, the $H_2O$ volume fraction will also decrease and the mixture temperature always remains above the dew point such that condensation within the plume is not possible. This is shown in the figure below which calculates plume temperatures, moisture content, and dew point temperature for the Bakken combustion products with air at 20°C and 60% relative humidity.

[Figure]

*Figure 1: Calculated (a) moisture content and (b) diluted plume and dew point temperatures for combustion of the Bakken flare mixture. Calculations conservatively assume an initial plume temperature (after an upper bound estimate of radiative cooling) of 1060°C with 16.5% $H_2O$ by volume (conservatively assuming complete combustion) diluted with lab air at 20°C and conservatively high relative humidity of 60% over a range of dilution factors. In all cases the mixture temperature remains well above the dew point temperature. At a dilution factor of 1000, the mixture temperature is 37°C and the dew point is roughly 10°C.*

*lines 113-114: Could the "<1 mrad beam steering that is interpreted as additional BC" be quantified in terms of measured BC? Or possibly as a fraction of the average measurement? It is hard for a non-specialist in this technique to interpret the magnitude of this error.*

Figure A3 (Appendix A.2) shows BC-equivalent values for the beam steering (which is shown in the Monte Carlo analysis to be unimportant relative to the measured variations in the species ratio). Lines 113-114 now close with a reference to this Appendix section.

*lines 128: Was the BC also averaged over 5 ms?*

Yes, thank you, the 1 MHz BC data were similarly averaged in 5-ms intervals to produce synchronous estimates of the BC and $H_2O$ volume fractions. We have added the following line to the end of the paragraph clarifying this point:

> *The 1654 nm laser was held at a constant wavelength, tuned away from absorbing gas species, and the 1-MHz data were averaged over 5-ms periods to produce synchronous BC and $H_2O$ volume fraction estimates.*

*line 155: BC or soot volume fraction is a niche term. Why not use mass concentration?*

We might argue that volume fraction is a common term when talking about combustion sources as opposed to ambient concentrations, but more importantly the measured BC volume fraction does not require an assumption about BC density and produces a dimensionless ratio with the $H_2O$ volume fraction, as measured by the spectroscopic technique. Conversely, if we assumed a BC density and reported a BC mass concentration, we would also want the mass concentration of $H_2O$ to produce a dimensionless ratio. However, computing the mass concentration of $H_2O$ from the spectral measurements would require knowledge of the local gas temperature. Although the presented optical system provides estimates of path-averaged temperature as part of the profile fit, incorporating this estimate back into the species ratio could introduce unnecessary uncertainty into the species ratio analysis.

*line 165: Using a back of the envelope calculation, it seems like this extinction coefficient is large. If the mass extinction cross section (m²g⁻¹) is around 4 and the particle density is 1.7 g cm⁻³, this value would be around 7x10⁶ m²m⁻³ (m⁻¹). Perhaps a more explicit explanation of the methodology used would clear up the confusion.*

We apologize that there was a typo on the exponent of the extinction ratio in the manuscript, which should have read $2.6 \times 10^6$ m $^2$m$^{-3}$ as has now been corrected in the revised text.

Using the reviewer's suggested BC density of 1.7 g cm$^{-3}$, our assumed extinction coefficient yields a mass extinction cross-section of 1.5 m$^2$g$^{-1}$ at 1654 nm and, for the scatter-to-absorption ratio of ~0.02 extrapolated from Migliorini et al. (2011), yields a mass absorption cross-section (MAC) of roughly equal magnitude. Assuming a typical absorption Ångström exponent (AAE) of unity (Conrad and Johnson, 2019), this would equate to a MAC value at 550 nm of 4.5 m$^2$m$^{-3}$ which compares well with the average value of 7.5 m$^2$m$^{-3}$ suggested by Bond and Bergstrom (2006).

*lines 167-175: What fraction of the light attenuation at the 1428 wavelength was estimated to be due to BC for a typical measurement?*

At 1428 nm, approximately 63% of the attenuation is attributable to $H_2O$ and 37% is due to BC. Specifically, considering the median estimate of BC for the aggregated Ecuador flare dataset, BC attenuates the signal by 1.4% whereas the $(2\nu_2+\nu_3)$ ro-vibrational peak of $H_2O$ at 7006.12 cm$^{-1}$ (1427.32 nm) attenuates the laser signal by 2.4%.

*line 241: What does "similarly averaged" mean? Was there also a wavelength sweep at 1654nm? Figure 4: "Sample point index" is not very meaningful. Could this be plotted with time on the x-axis instead? My understanding is that this is the time within the 5ms sweep.*

We have edited this line to clarify the fact that the 1654-nm signal is averaged as if it possessed a lead and lag sweep, like the 1428-nm signal, to ensure that the $H_2O$ spectra and BC attenuation values correspond in time. This then permits the BC attenuation to be isolated from the 1428-nm laser signal point-by-point. The sentence now reads,

> *The measurement from the 1654-nm laser shown in Fig.* **Error! Reference source not found.***b was averaged similarly to the 1428-nm (by averaging lead and lag) to allow mapping of the BC attenuation data onto the 1428 nm signal.*

The averaging of points to produce Fig. 4 means that each sample point corresponds to two points in the raw data (i.e. from lead and lag during the sweep). For this reason, we feel that it is more transparent to plot the sweep vs. index.

*line 250: How much of the data was omitted due to the SSE threshold?*

The SSE value was intentionally chosen to be restrictive so as to provide conservatively low estimates of the BC/$H_2O$ skewness and standard deviation as described in Appendix A.4. Thus, of the 30,000 sweeps of the $H_2O$ laser at each condition, roughly 70% of measurement results were filtered out by limiting the SSE. A less restrictive filter would only increase the skewness and variability of the final BC/$H_2O$ ratios as further discussed in Appendix A.4 and in Seymour (2019).

*lines 260-267: This paragraph feels like it belongs in an "experimental design" section of the methods, rather than in Results.*

As suggested, we have incorporated this information into Section 2.1 to augment the descriptions of the experimental program.

*line 271: A general correlation between the species is not expected "since the production of BC necessitates the production of H2O". Most H2O is produced through the generation of CO2. A tiny amount of BC is produced relative to H2O, thus the variably in this ratio is directly related to the variability in BC formation, which varies due to micro-physical changes in combustion conditions. Whereas, I suspect the variability observed in H2O is primarily due to dilution. It does not seem surprising, or particularly significant that such variability was observed.*

We believe our original wording was unnecessarily confusing. Given the existence of some fixed mean emission rate value for each species that is proportional to the overall rate of combustion, emitted species must always be correlated in the mean, irrespective of any variability in the instantaneous production rates of different species. In practice, for a steady

burning flame, correlations will primarily occur as bulk dilution of the turbulent plume with air simultaneously dilutes co-located species. Former line 271 (now line 264) explicitly addresses this point,

> *Although there is a general correlation between the species, as should be expected since the mean emission rate of each species is proportional to the overall rate of combustion, the results show considerable scatter about the central trend. Importantly, if BC/H2O ratios were fixed, then dilution of the turbulent plume with air would only populate points along the central line. The scatter is also not attributable to measurement noise which is roughly an order of magnitude less than the apparent variation.*

As more clearly stated in this revised text, the critical point is that although dilution of the plume with air would result in points along the central correlation line of Fig. 6, the measured data show significant scatter about this central trend that is well beyond measurement uncertainties. This variability is the first evidence in the paper that instantaneous BC/H2O ratios cannot be assumed constant.

*line 276-277: "..is not explained by a linear model..." Why should it be? Why isn't a large variance expected? Are we sure these $R^2$ values should be characterized as "low"?*

Please see previous response. Scatter about the central trend (beyond uncertainties) necessarily means that instantaneous species concentrations are not perfectly correlated and BC/H2O ratios are not constant. While a large variance in time-resolved species emission rates may be expected, dilution of the turbulent plume with air would be expected to similarly affect all species in a mixed plume such that simultaneously measured BC and H2O concentrations should only populate the central line. The $r^2$ values of 0.50 and 0.56 suggest that approximately half of the variation is unexplained by the linear model assuming a constant BC/H2O ratio. We have edited the sentence to make this more clear.

*line 286: Is there a theoretical reason why the ratios and variability were different between the two flare gas mixtures? This result seemed interesting enough to warrant inclusion of a figure, but the implications and interpretation are under-explored.*

Yes, this is a simple result of the difference in heating values between the two fuels. Studies such as McEwen and Johnson (2012) and Conrad and Johnson (2017) have demonstrated a direct relationship between fuel higher heating values and black carbon emission factors for flares. The increase in higher heating value between Bakken and Ecuador fuels (see Table 1 in the manuscript) results in a higher BC yield from flares of Ecuador fuel compositions. However, from the stoichiometric balance, the rate of H2O production is similar for the two fuels and therefore the mean BC/H2O ratio is higher for the Ecuador mixture. We have updated the manuscript to include this discussion.

Similarly, the increased variability in Ecuador BC/H2O estimates is associated with an increase in the total BC yield and thus a larger potential range of BC values. In Sect. 4, we discuss that although we observe an increase in the variability of the BC/H2O ratio of Ecuador fuels relative to Bakken fuels, the coefficient of variance (mean-normalized standard deviation) is quite similar, suggesting that the any perceived changes in BC/H2O variability are attributable to an increase in total BC presence.

*line 292-293: "...driven by turbulent dynamics..." I suspect that this is the main driver of variability, but it isn't clear to me how the comparable skewness of the distributions generated this conclusion.*

We have clarified this point as follows,

> *The consistency between the coefficient of variance and skewness for the two fuels shows that these species ratio variations do not scale with increased BC production and instead suggests that the uncorrelated variation in instantaneous species concentrations are most likely related to combustion dynamics and species mixing within the turbulent flame rather than fuel chemistry.*

*line 331: At first glance, the skewness observed here appears to be lower than found by Conrad & Johnson (2017), which would suggest that high BC events are somewhat correlated with higher rates of combustion. How do these measurements compare to those observations?*

Conrad and Johnson (2017) reported skewness of instantaneous BC emission rates [g/s] (quantified through a control surface defined across the field of view of their camera) whereas the present work investigates the variability in BC/H2O species ratios [-]. As noted in previous responses (see especially Specific Comment 4), these two parameters are distinct.

*line 349: After "ratio...", add "for that individual flare".*

Added as suggested.

*line 366: This description of "path-averaged" and "point-sampling" doesn't correctly describe the differences between the drone samples and the path-averaged optical measurements described here.*
*1) The drone measurements (as well as those conducted with airplanes) are sampled through the plume, collecting a volume of air for the time it takes to pass through the plume. The assertion that these samples capture less of the plume than a single optical measurement is not convincing; the volume captured in the optical path-averaged measure is tiny. 3) It is also likely that the plume, measured further downstream than here is much better mixed than from 2m from the flare, such that even small samples are likely to have less variability in ratios. 4) The description here also implies that these measures report a single measurement as if it was representative, but they do not. Reporting is made of averages of a number of passes. 5) The description of the path-average sampling seems to suggest that more of the plume is measured than with other sampling techniques, but the volume represented by a laser beam across the plume is small and also fails to capture most of the plume, which varies spatially as well as temporally.*

It is important to keep in mind that our optical system was not intended to replicate any specific field measurement technique. Instead, this instrument was developed to investigate variability in $BC/H_2O$ species ratios in turbulent plumes, which is a challenge to be overcome by all techniques involving explicit or implicit assumptions about the stability of species ratios in a turbulent plume. The presented system measured simultaneously averaged BC and $H_2O$ concentrations along an optical sample path transecting the turbulent plume and reveals strong variability in the measured species ratios with a skewed distribution. A bootstrap analysis procedure is then used to develop minimum sampling criteria based on the present experiments that is offered to others in the community as a starting point in planning future campaigns and potential implications are discussed.

We have edited Section 4.1 to make sure this is clear, where the second paragraph now opens with,

> *Although the presented optical instrument was not intended to replicate the measurements of any specific field technique, the experimental observations of variable $BC/H_2O$ ratio can still be used to infer general impacts on flare performance estimates from a limited dataset.*

At former line 366 (now line 371), we explained that point-sampling approaches (e.g. drones) should, in general, be subject to higher uncertainty from species ratio variability than path-averaged techniques. We have clarified this observation by removing the cited line and stating,

> *Different measurement techniques can be expected to be more or less susceptible to these issues, but in general shorter path measurements (e.g., extractive samples from fixed probes or stationary drones) are expected to observe greater variability in species correlations without the benefit of long path averages to smooth out relative species variations.*

As noted in the manuscript and the next comment below, turbulent mixing and diffusion may help as the plume moves downstream, but this is not necessarily true. Under crosswind conditions, plumes of flares can have two distinct regions with different composition and temperature profiles (Poudenx, 2000; Poudenx et al., 2004) that each evolve to become more dilute and more spatially distributed, further complicating measurements. Indeed, experiments (Johnson et al., 2001) and flow visualization (Johnson and Kostiuk, 2002) have shown that methane can be emitted in intermittent bursts beneath the wind-deflected flame and remain spatially distributed and thermally distinct from the warmer $CO_2$-rich plume. As noted in our response to Specific Comment 1, larger absolute sampling volumes can be expected to help reduce species ratio variability only if these larger volumes represent an equal or greater fraction of the plume volume at the downstream measurement location of the sample. However, the specifics of any technique are unique and, within the context of averaging over space and time, would depend on the sampling path (e.g. stationary, flying crosswind, upwind, etc.), scale of the plume at the sampling location, and sample duration at any one location. In addition to other edits in the revised Section 4.1, the paragraph in question starting at former line 366 now closes with,

> *This [species correlation] may be less of an issue in full-sized aircraft sampling approaches (e.g. Gvakharia et al., 2017; Weyant et al., 2016) which would be expected to measure a much larger volume than the presented technique; however, the required data averaging to obtain representative results will depend on the relative plume volumes sampled at that downstream measurement position, sampling duration, and how the technique interrogates the plume (e.g. stationary point or line-of-sight, transect downwind or crosswind, etc.).*

*line 375-379: Is methane leaking germane to the discussion about combustion pollutant ratios? The idea that measured variability in BC/H₂O extends to other species should be more grounded in combustion theory. I suspect the BC/H₂O ratios have little bearing on the release of unburnt fuel, but are fundamentally related to the relative emissions of CO₂.*

This text speaks directly to point 3 raised by the reviewer above in the comment referencing line 366. Specifically,

> *While turbulent plume motion could have beneficial mixing effects for some measurement approaches, this is not necessarily the case. High-frequency methane sampling measurements (Johnson et al., 2001) and flow visualization experiments (Johnson and Kostiuk, 2002) of flares in crossflow have also shown that methane can be emitted as intermittent bursts below the lee-side of a deflected flare flame, where these bursts are separated from the main CO₂-rich plume.*

These studies show how $CO_2$ can be spatially distinct form methane in a flare plume and how, in the context of the present results, we can thus expect local $CO_2/CH_4$ species ratios to be uncorrelated. We have edited the sentence following the quoted text to make this more clear,

> *These observations imply that CO₂ and unburnt fuel (methane) may be uncorrelated and that issues of species correlation extend beyond BC production, with an overall effect of further hindering field measurements.*

*line 415: Is the 150 cm the distance from one side of the optical hoop to the other (from collimator to wedged window in Figure 2)? If so, what are the nodes?*

The 150 cm distance is the distance from wedged window (launch side) to wedged window (collection side).

Line 415, part of Appendix A.1, is describing the used of large eddy simulation (LES) data to validate the analysis algorithm. The simulation domain had a 2.5-cm resolution which means that there were 60 nodes across the 150-cm optical path which contained the temperature and species volume fraction data. These data were used to test the ability of the analysis algorithm to correctly reproduce the known path-averaged $H_2O$ and BC concentrations from the LES data.

*line 434: How can you know the BC was "correctly estimated to within 0.02 ppb" without external calibration? Is this actually a measure of measurement uncertainty?*

Please refer to previous response to Specific Comment 3.

Appendix A.1 (which includes line 434) discusses performance testing of the measurement/analysis algorithms. Using the instantaneous concentration and temperature fields of a large eddy simulation (LES) of a natural gas flare in crossflow, simulated laser signals were generated (representing the path-integrated data that would be seen by the detectors) that were then intentionally corrupted with added Gaussian noise at levels greater than those seen in the experiment. The algorithms were demonstrated to be capable of correctly estimating the known path-average BC concentrations from the LES to within 0.02 ppb. Moreover, this uncertainty was entirely attributable to the simulated measurement noise. As discussed in Appendix A.2, during experiments the BC uncertainty was primarily driven by beam steering (with a similar Gaussian response) and was directly considered in the Monte Carlo simulations of Appendix A.5.

*Figure A1b: It isn't clear how features within the beam are determined. Were they measured or modeled?*

We apologize that there was a typo in the figure. "LOS data" has now been corrected to "LES data". We have also edited the caption to make it clear that large eddy simulation (LES) data are shown.

---

## Author Comment (AC2)

**Manuscript ID:** amt-2020-472
**Title:** Species Correlation Measurements in Turbulent Flare Plumes: Considerations for Field Measurements
**Authors:** Seymour, S.P.; Johnson, Matthew R.

**Point-by-point Response to Comments by Referee #2**

**General comments**

*This paper describes the application of a novel device to investigate the short-term variability of the BC/H2O ratio in a lab-scale flare using two different fuel compositions representative of the Bakken and Ecuador regions. The authors performed a Monte Carlo simulation to estimate variation and skewness in the BC/H2O ratio. They found that high ratios can be related to high BC production and further discuss the impact it could have on uncertainty and field measurements that assume a constant ratio.*

*The topic of this study is in the scope of the Atmospheric Measurement Techniques Chemistry journal and addresses an important subject relevant to field measurements of flaring emissions. Nevertheless, the implications of the study would benefit with more clarification and the Conclusions need to be improved since they read very similar to the Abstract.*

> We thank for your thoughtful input on our manuscript and your highlighting of the importance of this subject for emissions quantification. We have responded to your comments in detail below and have revised the conclusions to reduce overlap with the abstract as requested. Thank you for your help in improving the clarity of our manuscript.

**Specific comments**

1) *In Section 4.1 the authors mention the impact that crosswind might have on the short-term variability of the ratio. As part of the experimental method they also use data from CFD simulations of flares under crosswind conditions. However, the authors use a vertical flame to conduct the measurements of the BC/H2O ratio and in the manuscript it is not mentioned if they conducted experiments under crosswind conditions and why. My question is if the results obtained with a vertical flame can be representative also for crosswind conditions. What are the implications for the experimental device / setup to explore that? Are the histograms of both the sample mean and ratio distribution subject to change, and if so, to what extent?*

> Although crosswind experiments are not possible in the current facility, as discussed in Section 4.1, we expect the presented experimental results for vertical flares without crosswind represent the most conservative case for minimizing species variability effects. Under crossflow conditions the radial symmetry of the plume is lost and experiments (Poudenx, 2000) have shown that crosswinds can cause separation of gas phase species into different regions of the plume, with different local carbon conversion efficiencies. Further experiments (Johnson et al., 2001) and flow visualization (Johnson and Kostiuk, 2002) have also demonstrated how crosswinds can cause intermittent bursts of unburnt fuel (methane) to be emitted from the underside of the wind-deflected flame. These bursts of unburnt fuel are spatially separate from the $CO_2$-rich plume and are thermally distinct, suggesting that $CO_2$ and methane would be uncorrelated in the plume. Although these results speak to variations in gaseous species ratios in crossflow, it is reasonable to assume that they would only serve to further complicate measurements reliant on well-correlated species assumptions. However, looking toward future experiments, the presented device and methodology could be directly applied in future measurements in a wind tunnel or similar facility. This is presently being explored by close collaborators.

2) *Regarding the implications of the study, the authors suggest that based on the results of the BC/H2O ratio, aircraft-based estimates may be under-sampled and the estimated black carbon emission rate would be biased low. However, the techniques might not be comparable between each other. Factors like a large variability in the operating conditions which can impact the estimates substantially even from day to day [Conrad and Johnson, 2017], or the difference in the volume sampled between these approaches are not discussed in detail in the manuscript. In addition, the measurements in the paper were obtained with freshly emitted BC which might not be the case for those obtained with an aircraft, especially those sampled kilometers*

*downwind. A brief discussion considering differences between non-intrusive and extractive sampling techniques will add more clarity to the paper.*

We agree that different techniques will be more or less susceptible to species ratio variations, and have added further discussion to the implications section (4.1) to make this more clear. As noted in the text, our optical system was not intended to reproduce any specific aircraft or other measurement approach, but "to determine whether variations in species ratios in a turbulent plume could be detected, and if so, to examine how these might affect field measurements that rely on the assumption of a fixed ratio as part of a molar- or mass-balance to quantify pollutant emission rates or combustion/destruction efficiency." Ultimately, the results showing both strong variation and skewed distributions of path-averaged $BC/H_2O$ ratios do have important implications for a range of approaches. As noted in the revised text, "in general, shorter path measurements (e.g., extractive samples from fixed probes or stationary drones) are expected to observe greater variability in species correlations without the benefit of long path averages to smooth out relative species variations." Spatial coverage of the inhomogeneous plume (see Poudenx, 2000) is also important as elaborated in Sect. 4.1. The potential relevance of sample volume would depend on the relative portion of the plume being captured, where we note,

> *This may be less of an issue in full-sized aircraft sampling approaches (e.g., Gvakharia et al., 2017; Weyant et al., 2016) which would be expected to measure a much larger volume than the presented technique; however, in general, the required data averaging to obtain representative results will depend on the relative plume volumes sampled at that downstream measurement position, sampling duration, and how the technique interrogates the plume (e.g. stationary point or line-of-sight, transect downwind or crosswind, etc.).*

It is also important to distinguish between variable BC emission rates and variable $BC/H_2O$ ratios which can be quite different. The cited time-resolved measurements in Figures 1 and 2 of Conrad and Johnson (2017) show that mean and instantaneous BC emission rates [g/s] are highly variable. However, prior to the present work it was not clear from the literature whether BC would be well-correlated with other combustion products, or specifically if species ratios, e.g. $BC/H_2O$ or $BC/CO_2$, might still be stable. The results suggest that variability in species ratios (e.g. $BC/H_2O$) is an added factor to be considered (i.e. in addition to variability in raw BC emission rates), where both are important in designing measurement campaigns to maximize accuracy.

**Technical corrections**

*Ln 7. Briefly state why these measurements are important.*

We have added this sentence to the abstract,

> *Incomplete combustion from these processes results in emissions of black carbon, unburnt fuels (methane), $CO_2$, and other pollutants.*

*Ln 21. Replace 'should be easily avoidable'. Consider rephrasing.*

We have rephrased this to, "can be avoided."

*Ln 27. Consider rephrasing 'up 3% from 2018' to something like (just an example) 'up to 3% higher than in 2018'.*

This sentence now reads,

> *…estimated to be 150 billion $m^3$, an increase of 3% from 2018, …*

*Ln 28. Maybe mention that methane is also considered a Short Lived Climate Pollutant (SLCP). It is indirectly mentioned for Black Carbon on Ln 38.*

We have revised the sentence to read:

> *Of the pollutants produced by gas flaring, black carbon (BC; the carbonaceous, strongly light-absorbing component of particulate matter) and methane are both Short-Lived Climate Pollutants (SLCP) that, along with $CO_2$, constitute the three most climate warming pollutants in the atmosphere (e.g. Bond et al., 2013; IPCC, 2013; Jacobson, 2010; Ramanathan and Carmichael, 2008).*

*Ln 66. Change 'been study' to 'been a study'.*

Thank you for catching this typo.  The line now reads,

> *...no study has attempted to directly inspect this potential issue*

*Ln 77. This sentence is not clear. Are these gas mixtures representative for both off-shore and on-shore facilities worldwide, or just for specific regions? A brief sentence would be good to clarify and help introduce the oil regions on Ln 86.*

We have removed the reference to flare gas mixtures in this opening sentence and to improve clarity, the following sentences now include the composition-relevant text originally at line 86 followed by a new sentence explaining the selection of these mixtures considering the relationship between heating value on sooting propensity.

*Ln 78. remove 'in schematic'.*

Removed as part of previous edit.

*Ln 84-85. Consider moving this sentence to the previous paragraph and merge with Ln 81.*

Thank you for the suggestion.  We have merged the sentence into the previous paragraph as part of the previous two edits.

*Figure 1. Please increase the font size. It can be hard to read.*

We have increased the font size in Figure 1 as suggested.

*Ln 96. Consider changing 'well away' to 'far enough' or similar wording.*

Changed to, "…far away from the flame."

*Ln 190. It is mentioned that the profile fitting technique was implemented to laminar flames. Is it specific for laminar flames? What are the basis to use it for turbulent flames like those in the paper?*

No, the technique is not specific to laminar flames.  Although Liu et al. (2007) experimentally demonstrated the profile fitting technique on laminar flames measuring $H_2O$, the fitting approach was introduced more generally for nonuniform flow fields.  More recently, Grauer et al. (2018) successfully used the same profile fitting approach to measure heated, turbulent gas escaping a vertical nozzle.  Nevertheless, as detailed in Appendix A.1, as part of this paper we conducted extensive simulations of the technique using large eddy simulation data, where we directly tested the ability of the algorithm to correctly reproduce known path-averaged temperature and $H_2O$ concentrations.  Results of this effort further proved the applicability of the technique and informed the subsequent Monte Carlo uncertainty analysis.

*Ln 274. Why is the error in H2O volume fraction much higher than the error in BC volume fraction?*

The $H_2O$ measurement is derived from the profile fitting technique, in which absorption measurements across two $H_2O$ absorption lines are used to simultaneously quantify the path-averaged $H_2O$ volume fraction and temperature distributions (see previous response and Sect. 3.3).  BC volume fraction, however, is directly measurable from the attenuation signal on the 1654 nm laser.  The difference in uncertainties is a direct result of the added steps required for the $H_2O$ measurement.

*Ln 286. "an apparent dependence on flare gas composition." It is an interesting result but it was only mentioned and not discussed with more detail. Did you consider the effect of the composition on soot chemistry ? According to Table 1, Ecuador composition has more branched-chain isomers. It could produce different intermediate radicals, which affect the formation of molecular soot precursors [Lei et al., 2020].*

Thank you for this suggestion.  Studies by McEwen and Johnson (2012) and Conrad and Johnson (2017) have demonstrated a relationship between fuel higher heating values and black carbon yield.  In our experiment, we would expect the higher BC production observed from the Ecuador mixture because of its larger heating value (see Table 1 in

the manuscript). Although the Ecuador mixture produces a higher BC yield, its production of $H_2O$ is roughly the same as the Bakken mixture, resulting in an increase $BC/H_2O$ ratio. We have updated the manuscript to include this discussion.

*Ln 705. Figure 2. Change 'shown' to 'show'.*

Typo corrected, thank you.

*Figure 3. include the meaning of qamb/Tamb, etc also in the Figures' description.*

The figure caption now includes relevant definitions. Thank you for pointing this out.

*Figure 5. Spell out 'NM' in the Figure's description.*

Spelled out "Nelder-Mead" as suggested.

*Ln 291. Remove the hyphen, it might be read as a negative value.*

Removed as suggested.

*Ln 340. remove 'etc.'*

Removed as suggested.

**References**

Bond, T. C., Doherty, S. J., Fahey, D. W., Forster, P. M., Berntsen, T. K., DeAngelo, B. J., Flanner, M. G., Ghan, S., Kärcher, B., Koch, D., Kinne, S., Kondo, Y., Quinn, P. K., Sarofim, M. C., Schultz, M. G., Schulz, M., Venkataraman, C., Zhang, H., Zhang, S., Bellouin, N., Guttikunda, S. K., Hopke, P. K., Jacobson, M. Z., Kaiser, J. W., Klimont, Z., Lohmann, U., Schwarz, J. P., Shindell, D. T., Storelvmo, T., Warren, S. G. and Zender, C. S.: Bounding the role of black carbon in the climate system: A scientific assessment, J. Geophys. Res. Atmos., 118(11), 5380–5552, doi:10.1002/jgrd.50171, 2013.

Caulton, D. R., Shepson, P. B., Cambaliza, M. O. L., McCabe, D., Baum, E. and Stirm, B. H.: Methane destruction efficiency of natural gas flares associated with shale formation wells., Environ. Sci. Technol., 48(16), 9548–54, doi:10.1021/es500511w, 2014.

Conrad, B. M. and Johnson, M. R.: Field measurements of black carbon yields from gas flaring, Environ. Sci. Technol., 51(3), 1893–1900, doi:10.1021/acs.est.6b03690, 2017.

Grauer, S. J., Conrad, B. M., Miguel, R. B. and Daun, K. J.: Gaussian Model for Emission Rate Measurement of a Heated Plume using Hyperspectral Data, J. Quant. Spectrosc. Radiat. Transf., 206, 125–134, doi:10.1016/j.jqsrt.2017.11.005, 2018.

Gvakharia, A., Kort, E. A., Brandt, A. R., Peischl, J., Ryerson, T. B., Schwarz, J. P., Smith, M. L. and Sweeney, C.: Methane, black carbon, and ethane emissions from natural gas flares in the Bakken Shale, North Dakota, Environ. Sci. Technol., 51(9), 5317–5325, doi:10.1021/acs.est.6b05183, 2017.

IPCC: Chapter 8: Anthropogenic and Natural Radiative Forcing, in Climate Change 2013: The Physical Science Basis. Contribution of Working Group I to the Fifth Assessment Report of the Intergovernmental Panel on Climate Change, edited by T. F. Stocker, D. Qin, G.-K. Plattner, M. Tignor, S. K. Allen, J. Boschung, A. Nauels, Y. Xia, V. Bex, and P. M. Midgley, pp. 659–740, Cambridge University Press, Cambridge, United Kingdom and New York, NY, USA., 2013.

Jacobson, M. Z.: Short-term effects of controlling fossil-fuel soot, biofuel soot and gases, and methane on climate, Arctic ice, and air pollution health, J. Geophys. Res., 115(D14209), 1–24, doi:10.1029/2009JD013795, 2010.

Johnson, M. R. and Kostiuk, L. W.: Visualization of the fuel stripping mechanism for wake-stabilized diffusion flames in a crossflow, in IUTAM Symposium on Turbulent Mixing and Combustion, vol. 70, edited by A. Pollard and S. Candel, pp. 295–303, Springer Netherlands, Dordrecht., 2002.

Johnson, M. R., Wilson, D. J. and Kostiuk, L. W.: A fuel stripping mechanism for wake-stabilized jet diffusion flames in crossflow, Combust. Sci. Technol., 169(1), 155–174, doi:10.1080/00102200108907844, 2001.

Liu, X., Jeffries, J. B. and Hanson, R. K.: Measurement of Non-Uniform Temperature Distributions Using Line-of-Sight

Absorption Spectroscopy, AIAA J., 45(2), 411–419, doi:10.2514/1.26708, 2007.

McEwen, J. D. N. and Johnson, M. R.: Black carbon particulate matter emission factors for buoyancy-driven associated gas flares, J. Air Waste Manage. Assoc., 62(3), 307–321, doi:10.1080/10473289.2011.650040, 2012.

Poudenx, P.: Plume sampling of a flare in crosswind: structure and combustion efficiency, M.Sc. Thesis, University of Alberta, Edmonton, AB, Canada, Edmonton., 2000.

Ramanathan, V. and Carmichael, G.: Global and regional climate changes due to black carbon, Nat. Geosci., 1(4), 221–227, doi:10.1038/ngeo156, 2008.

U.S. EPA: Report to Congress on Black Carbon, United States Environmental Protection Agency (U.S. EPA), Research Triangle Park, NC., 2012.

Weyant, C. L., Shepson, P. B., Subramanian, R., Cambaliza, M. O. L. L., Heimburger, A., Mccabe, D., Baum, E., Stirm, B. H. and Bond, T. C.: Black carbon emissions from associated natural gas flaring, Environ. Sci. Technol., 50(4), 2075–2081, doi:10.1021/acs.est.5b04712, 2016.

---

## Referee Report (RR1)

Review for "Species Correlation Measurements in Turbulent Flare Plumes: Considerations for Field Measurements". Second Iteration.

The authors addressed all the specific comments and technical corrections by the reviewers and improved the manuscript accordingly. Some of their answers were supported with additional graphics and discussion. They included arguments in their response to clarify between their experimental technique and other experimental techniques (aircraft-based mainly), giving more context to their results.

An additional brief paragraph about the variability of BC/H2O and BC/CO2 ratios was included, and they justify the development and main features of optical attenuation measurements to support the experimental technique. In addition, a brief paragraph related to crosswind conditions was included in the manuscript.

I recommend publication of the manuscript after addressing the following minor technical corrections.

**Technical corrections**

Figure 3. Please correct the Figure. qamb appears twice, and qcol is missing
Figure 8. Perhaps add "*aggregated*" in the Figure's description to be in correspondence with the wording in Table 2.

Ln 219. Change "*meant*" to "means"
Ln 362. Change "*minimum required samples size*" to "minimum sample sizes"
Ln 383. Consider changing "*interrogates*" for another word
Ln 384. Change "*etc*" to something like "among others"
Ln 401. Consider removing "*perfectly-*"

---

## Author Response (AR2)

**Manuscript ID:** amt-2020-472
**Title:** Species Correlation Measurements in Turbulent Flare Plumes: Considerations for Field Measurements
**Authors:** Seymour, S.P.; Johnson, Matthew R.

**Response to Comments by Referee**

*The authors addressed all the specific comments and technical corrections by the reviewers and improved the manuscript accordingly. Some of their answers were supported with additional graphics and discussion. They included arguments in their response to clarify between their experimental technique and other experimental techniques (aircraft-based mainly), giving more context to their results.*

*An additional brief paragraph about the variability of BC/H2O and BC/CO2 ratios was included, and they justify the development and main features of optical attenuation measurements to support the experimental technique. In addition, a brief paragraph related to crosswind conditions was included in the manuscript.*

*I recommend publication of the manuscript after addressing the following minor technical corrections.*

We thank the referee for their review of our manuscript and their acknowledgement of our strengthened discussion following comments from the reviewers. We have implemented the remaining technical corrections, as suggested.

*Technical corrections*

*Figure 3. Please correct the Figure. qamb appears twice, and qcol is missing*

*Figure 8. Perhaps add "aggregated" in the Figure's description to be in correspondence with the wording in Table 2.*

*Ln 219. Change "meant" to "means"*

*Ln 362. Change "minimum required samples size" to "minimum sample sizes"*

*Ln 383. Consider changing "interrogates" for another word*

*Ln 384. Change "etc" to something like "among others"*

*Ln 401. Consider removing "perfectly-"*